# RECOVERY OF CAUSAL GRAPH INVOLVING LATENT VARIABLES VIA HOMOLOGOUS SURROGATES

**Xiu-Chuan Li    Jun Wang    Tongliang Liu**[*]
Sydney AI Centre, University of Sydney

## ABSTRACT

Causal discovery with latent variables is an important and challenging problem. To identify latent variables and infer their causal relations, most existing works rely on the assumption that latent variables have pure children. Considering that this assumption is potentially restrictive in practice and not strictly necessary in theory, in this paper, by introducing the concept of homologous surrogate, we eliminate the need for pure children in the context of causal discovery with latent variables. The homologous surrogate fundamentally differs from the pure child in the sense that the latter is characterized by having strictly restricted parents while the former allows for much more flexible parents. We formulate two assumptions involving homologous surrogates and develop theoretical results under each assumption. Under the weaker assumption, our theoretical results imply that we can determine each variable's ancestors, that is, partially recover the causal graph. The stronger assumption further enables us to determine each variable's parents exactly, that is, fully recover the causal graph. Building on these theoretical results, we derive an algorithm that fully leverages the properties of homologous surrogates for causal graph recovery. Also, we validate its efficacy through experiments. Our work broadens the applicability of causal discovery. Our code is available at: https://github.com/XiuchuanLi/ICLR2025-CDHS

## 1 INTRODUCTION

Causality is a basic concept in natural and social sciences, playing a pivotal role in explanation, prediction, decision making and control (Zhang et al., 2018). In particular, causality has driven significant progress in machine learning (Yao et al., 2021; 2023; Huang et al., 2023; Hong et al., 2024; Lin et al., 2024; Sun et al., 2024). The gold standard for uncovering causality is conducting randomized experiments, but this is usually prohibitively expensive and time-consuming. Consequently, researchers have increasingly turned to causal discovery, which aims to infer causal relations from observational data alone. Traditional causal discovery methods typically assume that all task-relevant variables have been enumerated and measured (Spirtes & Glymour, 1991; Chickering, 2002; Shimizu et al., 2006; Hoyer et al., 2009; Zhang & Hyvärinen, 2009), but this assumption often fails in real-world scenarios, prompting the development of causal discovery with latent variables. These methods can be classified into three categories. The first category assumes that latent variables are mutually independent (Hoyer et al., 2008; Maeda & Shimizu, 2020; Yang et al., 2022; Cai et al., 2023). The second category allows the presence of causally-related latent variables but cannot identify latent variables, let alone their causal relations (Spirtes et al., 1995; Claassen et al., 2013; Claassen & Bucur, 2022). The third category not only allows the presence of causally-related latent variables but also can identify latent variables along with their causal relations (Silva et al., 2006; Xie et al., 2020; Huang et al., 2022; Jin et al., 2024). Our work belongs to the third category.

Recent works in the third category predominantly rely on the pure children assumption that latent variables have pure children. They not only identify latent variables by locating their pure children but also use their pure children as proxies to infer their causal relations. These works can be further categorized into two groups. Some works (Silva et al., 2006; Shimizu et al., 2009; Kummerfeld & Ramsey, 2016; Cai et al., 2019; Chen et al., 2022; Zeng et al., 2021; Xie et al., 2022; Chen et al., 2023) make the special pure children assumption that each latent variable has multiple pure children.

---

[*]Correspondence to Tongliang Liu (tongliang.liu@sydney.edu.au).

Here, a variable is said a pure child of anther only if the latter is the only parent of the former. Other works (Xie et al., 2020; 2024; Huang et al., 2022; Dong et al., 2024; Jin et al., 2024) make the general pure children assumption that each latent variable belongs to a latent set (comprising one or more latent variables) which has sufficient pure children. Here, a variable is said a pure child of a latent set only if all parents of the former are within the latter. The general pure children assumption is often accompanied by local unidentifiability: for multiple latent variables within a latent set, even the existence (let alone directions) of the causal relations between them might be undeterminable. In summary, the concept of pure child is characterized by having strictly restricted parents.

Although the pure children assumption is widely employed for the sake of tractability, Adams et al. (2021) argue that it is restrictive in practice and prove that it is not necessary for identifiability of linear non-Gaussian acyclic models with latent variables. In this paper, by introducing the concept of homologous surrogate, we eliminate the need for pure children. The homologous surrogate fundamentally differs from the pure child in the sense that the former allows for much more flexible parents. Specifically, if an observed variable $O$ is a pure child of a latent variable $L$, $O$ must have only one parent $L$; but if $O$ is a homologous surrogate of $L$, $O$ is allowed (not mandatory) to have other parents besides $L$, such as other latent parents provided that they are all $L$'s ancestors. Taking $O_3$ in Fig. 1 as an example, it has two parents $L_1$, $L_2$ where $L_1$ an ancestor of $L_2$. Correspondingly, it is a homologous surrogate but not a pure child of $L_2$. It is common that an observed variable is a homologous surrogate but not a pure child of a latent variable. For instance, a company's stock price is caused by both its intrinsic value and macroeconomic environment, and macroeconomic environment also impacts its intrinsic value, so its stock price is a homologous surrogate but not a pure child of its intrinsic value. Besides, a boy's violent behavior is caused by both his violent tendency and the parental supervision, and the parental supervision also impacts his violent tendency, so his violent behavior is a homologous surrogate but not a pure child of his violent tendency.

We begin with the assumption that each latent variable has at least one (rather than multiple) homologous surrogate. Under this assumption, we develop theoretical results implying that the causal graph can be partially recovered, that is, we can determine whether any variable is an ancestor of any other variable. From these theoretical results, we derive an algorithm that sequentially identifies latent variables by locating their homologous surrogates, progressing from roots to leaves, during which process the causal graph is also partially recovered. We then develop further theoretical results with an extra as-

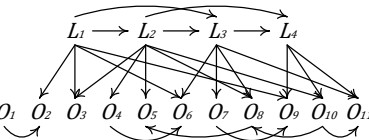

Figure 1: A causal graph that can be fully recovered by our algorithm. $L_i$ refers to latent variables while $O_i$ refers to observed variables.

sumption that if a latent variable is an ancestor of another, the latter must have two generalized homologous surrogates that are not children of the former, where the generalized homologous surrogate is a variant of the homologous surrogate but is subject to fewer restrictions. These theoretical results imply that the causal graph can be fully recovered, that is, we can determine whether any variable is a parent of any other variable, from which we also derive an algorithm. Building on the partial recovery result, this algorithm first locates latent variables' generalized homologous surrogates, then uses them to infer the causal relations between latent variables. Combining this information with the partial recovery result, the causal graph can be fully recovered. A causal graph that can be fully recovered by our algorithm is shown as Fig. 1, where no latent variable has multiple pure children.

The major innovations of our work are summarized as follows.

- We introduce a new concept called homologous surrogate to facilitate causal discovery with latent variables, which fundamentally differs from the pure child. While the latter is characterized by having strictly restricted parents, the former allows for much more flexible parents.

- We formulate two assumptions involving homologous surrogates and develop novel theoretical results under each assumption. These theoretical results imply that the causal graph can be partially/fully recovered under the weaker/stronger assumption.

- Building on our theoretical results, we derive a systematic and innovative algorithm which fully leverages the properties of homologous surrogates for causal graph recovery. We demonstrate the efficacy of our algorithm through experiments.

In summary, our work broadens the applicability of causal discovery. It may not only inspire further research in this direction but also benefit research in natural and social sciences. Due to space limit, we defer detailed discussion on related works to App. A.

## 2 PRELIMINARY

In this paper, we focus on the linear non-Gaussian acyclic model (LiNGAM) with latent variables whose causal graph $\mathcal{G} = (\mathbf{V}, \mathbf{E})$ is a directed acyclic graph (DAG). $\mathbf{V} = \mathbf{L} \cup \mathbf{O}$ where $\mathbf{L}$ and $\mathbf{O}$ respectively denote the set of latent and observed variables. Each variable in $\mathbf{L}$ and $\mathbf{O}$ follows

$$L_i = \sum_{L_j \in \mathbf{L}} a_{L_i}^{L_j} L_j + \epsilon_{L_i}, \quad O_i = \sum_{L_j \in \mathbf{L}} a_{O_i}^{L_j} L_j + \sum_{O_j \in \mathbf{O}} a_{O_i}^{O_j} O_j + \epsilon_{O_i}. \tag{1}$$

$a_{V_j}^{V_i}$ denotes the direct causal strength from $V_i$ to $V_j$, $a_{V_j}^{V_i} \neq 0$ if and only if $V_i$ is a parent of $V_j$. Given $V \in \mathbf{V}$, we denote its parents, children, neighbors, ancestors, and descendants by $\mathrm{Pa}(V), \mathrm{Ch}(V), \mathrm{Ne}(V), \mathrm{An}(V)$, and $\mathrm{De}(V)$. Particularly, a variable's ancestors/descendants do not include itself. In the following, for any $\mathbf{V}' \subseteq \mathbf{V}$, we abbreviate $\cup_{V \in \mathbf{V}'} \mathrm{Pa}(V)$ to $\mathrm{Pa}(\mathbf{V}')$. The latent parents and observed parents of $V$ are denoted by $\mathrm{Pa}_{\mathbf{L}}(V)$ and $\mathrm{Pa}_{\mathbf{O}}(V)$ respectively. $\epsilon_V$ refers to the exogenous noise of $V$, all exogenous noises have non-Gaussian distributions and are independent of each other. Without loss of generality, we assume that each exogenous noise has zero mean and the exogenous noise of each latent variable has unit variance.

We can rewrite Eq. (1) in a matrix form as

$$\begin{bmatrix} \mathbf{L} \\ \mathbf{O} \end{bmatrix} = \mathbf{A} \begin{bmatrix} \mathbf{L} \\ \mathbf{O} \end{bmatrix} + \begin{bmatrix} \epsilon_{\mathbf{L}} \\ \epsilon_{\mathbf{O}} \end{bmatrix}, \tag{2}$$

where

$$\mathbf{A} := \begin{bmatrix} \mathbf{A}_{\mathbf{L}}^{\mathbf{L}} & \mathbf{0} \\ \mathbf{A}_{\mathbf{O}}^{\mathbf{L}} & \mathbf{A}_{\mathbf{O}}^{\mathbf{O}} \end{bmatrix} \tag{3}$$

is the *adjacency matrix*. For $\mathbf{V}_1, \mathbf{V}_2 \subseteq \mathbf{V}$, $\mathbf{A}_{\mathbf{V}_2}^{\mathbf{V}_1}$ refers to the adjacent matrix from $\mathbf{V}_1$ to $\mathbf{V}_2$. Since $\mathbf{I} - \mathbf{A}$ is invertible (Shimizu et al., 2006), we can further rewrite Eq. (3) as

$$\begin{bmatrix} \mathbf{L} \\ \mathbf{O} \end{bmatrix} = \mathbf{M} \begin{bmatrix} \epsilon_{\mathbf{L}} \\ \epsilon_{\mathbf{O}} \end{bmatrix}, \tag{4}$$

where

$$\mathbf{M} = (\mathbf{I} - \mathbf{A})^{-1} = \begin{bmatrix} (\mathbf{I} - \mathbf{A}_{\mathbf{L}}^{\mathbf{L}})^{-1} & \mathbf{0} \\ (\mathbf{I} - \mathbf{A}_{\mathbf{O}}^{\mathbf{O}})^{-1} \mathbf{A}_{\mathbf{O}}^{\mathbf{L}} (\mathbf{I} - \mathbf{A}_{\mathbf{L}}^{\mathbf{L}})^{-1} & (\mathbf{I} - \mathbf{A}_{\mathbf{O}}^{\mathbf{O}})^{-1} \end{bmatrix} := \begin{bmatrix} \mathbf{M}_{\mathbf{L}}^{\mathbf{L}} & \mathbf{0} \\ \mathbf{M}_{\mathbf{O}}^{\mathbf{L}} & \mathbf{M}_{\mathbf{O}}^{\mathbf{O}} \end{bmatrix} \tag{5}$$

is the *mixing matrix* whose elements are called *mixing coefficients*. By convention, we assume the distribution over $\mathbf{V}$ is Markov and faithful to $\mathcal{G}$, which implies that $m_{V_j}^{V_i} \neq 0$ if and only if $V_i$ is an ancestor of $V_j$ or $V_i = V_j$. In the latter case, $m_{V_j}^{V_i} = 1$.

## 3 PARTIAL RECOVERY

**Definition 1.** *(homologous surrogate)* $O \in \mathbf{O}$ *is called a homologous surrogate of* $L \in \mathbf{L}$, *denoted by* $O \in \mathrm{HSu}(L)$, *if* $O \in \mathrm{Ch}(L)$, $\mathrm{Ch}(O) = \emptyset$, $\mathrm{An}_{\mathbf{L}}(O) = \mathrm{An}(L) \cup \{L\}$ *and* $\mathrm{An}_{\mathbf{O}}(O) \cap \mathrm{De}_{\mathbf{O}}(L) = \emptyset$.

**Example.** In Fig. 1, $\mathrm{HSu}(L_1) = \{O_2\}$, $\mathrm{HSu}(L_2) = \{O_3\}$, $\mathrm{HSu}(L_3) = \{O_6\}$, $\mathrm{HSu}(L_4) = \{O_9\}$.

**Remark.** Given $O \in \mathbf{O}$ and $L \in \mathbf{L}$, we detail the the connections and differences between "$O$ is $L$'s pure child" and "$O$ is $L$'s homologous surrogate" in the following. There is a consensus among previous works that if an observed variable $O$ is a pure child of a latent variable $L$, then $O$ must have no other parent except $L$. In this case, it is obvious that $O \in \mathrm{Ch}(L)$, $\mathrm{An}_{\mathbf{L}}(O) = \mathrm{An}(L) \cup \{L\}$ and $\mathrm{An}_{\mathbf{O}}(O) \cap \mathrm{De}_{\mathbf{O}}(L) = \emptyset$, that is, 3 out of 4 conditions in Def. 1 are satisfied. On this basis, some studies (Silva et al., 2006; Kummerfeld & Ramsey, 2016; Xie et al., 2023; Li et al., 2024) explicitly require that $O$ has no child while others (Shimizu et al., 2009; Cai et al., 2019; Xie et al., 2020; 2022; Huang et al., 2022; Chen et al., 2023) directly assume that there exists no edge between observed variables. That is, the remaining condition $\mathrm{Ch}(O) = \emptyset$ in Def. 1 is satisfied. $O$ is allowed to have children of its own only in very few works (Dong et al., 2024; Jin et al., 2024). Therefore, if $O$ is a pure child of $L$ in most senses, then $O$ is also a homologous surrogate of $L$. The reverse is not necessarily true because $L$'s homologous surrogate is allowed to have other parents.

**Intuition.** Suppose $O$ is $L$'s homologous surrogate. If $L$ is a root variable, then $O$ has no child and only one latent parent $L$. Otherwise, with both $L$'s ancestors and observed variables whose latent ancestors are a subset of $L$'s ancestors removed, $L$ becomes a root variable and $O$ still has no child and only one latent parent $L$. Because of this, homologous surrogates can be located from observed variables.

**Assumption 1.** $\forall L \in \mathbf{L}$, $\mathrm{HSu}(L) \neq \emptyset$ and $|\mathrm{Ch}(L)| \geq 2$.

**Example.** The causal graph shown as Fig. 1 satisfies Asmp. 1.

**Remark.** In previous works making the pure children assumption, the number of latent variables must be strictly smaller than their pure children. Instead, we only require one homologous surrogate per latent variable.

**Intuition.** Assuming that each latent variable has a homologous surrogate, latent variables can be sequentially identified by locating their homologous surrogates, progressing from roots to leaves, during which process the causal graph is also partially recovered. Also, we assume each latent variable has multiple children, otherwise it can be modeled as a noise (Silva et al., 2006).

**§ High-level Overview.** First, we identify observed root variables (Thm. 1), estimate their effects on others (Cor. 1), and then remove them (Cor. 2). Second, we identify latent root variables (Thm. 2, Props. 1 and 2), estimate their effects on others (Cor. 3), and then remove them (Cor. 4). Repeating these two procedures until all observed variables are removed, we can identify all latent variables and partially recover the causal graph. During this process, all operations on latent variables are implemented through their homologous surrogates.

**§ Initialization.** We denoted the set of removed variables by $\mathbf{J} \cup \mathbf{K}$ where $\mathbf{J} \subseteq \mathbf{L}, \mathbf{K} \subseteq \mathbf{O}$. In addition, for each $O_i \in \mathbf{O} \backslash \mathbf{K}$, there is an auxiliary variable $\tilde{O}_i$ which is a linear combination of $O_i$ and variables in $\mathbf{K}$ where the coefficient of $O_i$ is always 1 while that of each variable in $\mathbf{K}$ is not fixed. Initially, we let $\mathbf{J} = \mathbf{K} = \emptyset$, so $\tilde{O}_i = O_i$ for each $O_i \in \mathbf{O}$, it is trivial that Cond. 1 is valid.

**Condition 1.** *(1) For each $V \in \mathbf{V} \backslash (\mathbf{J} \cup \mathbf{K})$, $\mathrm{De}(V) \cap (\mathbf{J} \cup \mathbf{K}) = \emptyset$. (2) For each $L \in \mathbf{J}$ and $O \in \mathbf{K}$ where $\mathrm{Ch}(O) \neq \emptyset$, $m_{\tilde{O}_i}^L = m_{\tilde{O}_i}^O = 0$.*

**§ Identifying Observed Root Variables.** This can be accomplished based on Thm. 1.

**Definition 2.** *(Pseudo-residual (Cai et al., 2019)) Given three variables $V_1, V_2, V_3$ s.t. $\mathrm{Cov}(V_2, V_3) \neq 0$, the pseudo-residual of $V_1, V_2$ relative to $V_3$ is defined as*

$$\mathrm{R}(V_1, V_2 | V_3) = V_1 - \frac{\mathrm{Cov}(V_1, V_3)}{\mathrm{Cov}(V_2, V_3)} V_2. \tag{6}$$

**Intuition.** Pseudo-residual is a simple variant of the conventional residual. The former reduces to the latter when $V_2 = V_3$. Before Cai et al. (2019), similar concepts have already been used by earlier works (Drton & Richardson, 2004; Chen et al., 2017).

**Theorem 1.** *Suppose $O_i \in \mathbf{O} \backslash \mathbf{K}$, then $\mathrm{An}(O_i) \subseteq (\mathbf{J} \cup \mathbf{K})$ if and only if $\forall O_j \in \mathbf{O} \backslash (\mathbf{K} \cup \{O_i\})$, $\mathrm{R}(O_j, O_i | \tilde{O}_i) \perp\!\!\!\perp \tilde{O}_i$.*

**Intuition.** The part before "if and only if" means that all ancestors of $O_i$ are in $\mathbf{J} \cup \mathbf{K}$, that is, $O_i$ is a root variable among $\mathbf{V} \backslash (\mathbf{J} \cup \mathbf{K})$; the part after "if and only if" means that $O_i$ satisfies certain independence constraints. Therefore, this theorem provides a method for identifying observed root variables via statistical analysis.

**Example.** Suppose the underlying causal graph is shown as Fig. 1. Initially, $\mathbf{J} = \mathbf{K} = \emptyset$. We can identify $O_1$ as an observed root because $\forall O_j \in \{O_2, ..., O_{11}\}, \mathrm{R}(O_j, O_1 | \tilde{O}_1) \perp\!\!\!\perp \tilde{O}_1$.

**§ Estimating the Effects of Observed Root Variables.** This can be accomplished based on Cor. 1.

**Corollary 1.** *Suppose $O_i$ satisfies Thm. 1, then $\forall O_j \in \mathbf{O} \backslash (\mathbf{K} \cup \{O_i\})$, $m_{O_j}^{O_i} = \frac{\mathrm{Cov}(\tilde{O}_i, O_j)}{\mathrm{Cov}(\tilde{O}_i, O_i)}$.*

**Remark.** With the faithfulness assumption, $O_j$ is a descendant of $O_i$ if and only if $m_{O_j}^{O_i} \neq 0$.

**§ Removing Observed Root Variables.** This can be accomplished based on Cor. 2.

**Corollary 2.** *Suppose $O_i$ satisfies Thm. 1, if we update $\mathbf{K}$ to $\mathbf{K} \cup \{O_i\}$ and $\tilde{O}_j$ to $\tilde{O}_j - m_{O_j}^{O_i} \tilde{O}_i$ for each $O_j \in \mathbf{O}\backslash\mathbf{K}$, Cond. 1 is still valid.*

**Remark.** With observed root variables removed, some observed non-root variables before removal might become roots after removal, so we need to repeat the above three steps until there is no observed root variable, that is, no observed variable satisfies Thm. 1.

**§ Identifying Latent Root Variables.** We identify latent root variables by locating their respective homologous surrogates from observed variables. However, this cannot be achieved in a single step. Instead, we first locate observed variables that might be homologous surrogates (called candidate homologous surrogates) of latent root variables (Thm. 2), then check whether any two of them share a common latent parent (Prop. 1), and finally find true homologous surrogates (Prop. 2).

**Theorem 2.** *Suppose $\forall O \in \mathbf{O}\backslash\mathbf{K}, \text{An}(O) \not\subseteq \mathbf{J} \cup \mathbf{K}$. Given $O_i \in \mathbf{O}\backslash\mathbf{K}$, then $\text{Ch}(O_i) = \emptyset$, $\text{Pa}_\mathbf{O}(O_i)\backslash\mathbf{K} = \emptyset$, $|\text{Pa}_\mathbf{L}(O_i)\backslash\mathbf{J}| = 1$, and $\text{An}(\text{Pa}_\mathbf{L}(O_i)\backslash\mathbf{J}) \subseteq \mathbf{J}$ if and only if $\forall\{O_j, O_k\} \subseteq \mathbf{O}\backslash(\mathbf{K} \cup \{O_i\})$ where $\text{Cov}(\tilde{O}_i, O_j)\text{Cov}(\tilde{O}_i, O_k) \neq 0$, $\text{R}(O_j, O_k|\tilde{O}_i) \perp\!\!\!\perp \tilde{O}_i$.*

**Intuition.** The opening supposition requires that no observed variable in $\mathbf{O}\backslash\mathbf{K}$ is a root variable among $\mathbf{V}\backslash(\mathbf{J} \cup \mathbf{K})$, which can be ensured by iterating the aforementioned three steps involving observed root variables until there is no observed variable in $\mathbf{O}\backslash\mathbf{K}$ satisfying Thm. 1. In the subsequent statement, the part before "if and only if" means that $O_i$ has no child, no observed parent in $\mathbf{O}\backslash\mathbf{K}$, and only one latent parent in $\mathbf{L}\backslash\mathbf{J}$ which is exactly a root variable among $\mathbf{V}\backslash(\mathbf{J}\cup\mathbf{K})$. This is a necessary but not sufficient condition for $O_i$ being a homologous surrogate of a latent root variable among $\mathbf{V}\backslash(\mathbf{J}\cup\mathbf{K})$ (further explained in Remark later); the part after "if and only if" means that $O_i$ satisfies certain independence constraints. Therefore, this theorem provides a method for identifying candidate homologous surrogates of latent root variables via statistical analysis.

**Example.** Suppose the underlying causal graph is shown as Fig. 1. After removing the observed root variable $O_1$ based on Thm. 1, $\mathbf{J} = \emptyset, \mathbf{K} = \{O_1\}$. At this point, $\forall O \in \{O_2, ..., O_{11}\}, \text{An}(O) \not\subseteq \{O_1\}$. Then we can identify $O_2$ as a candidate homologous surrogate of a latent root variable because $\forall\{O_j, O_k\} \in \{O_3, ..., O_{11}\}, \text{R}(O_j, O_k|\tilde{O}_2) \perp\!\!\!\perp \tilde{O}_2$.

**Remark.** Based on Def. 1, it is trivial that the part before "if and only if" is a necessary condition for $O_i$ being a homologous surrogate of a latent root variable among $\mathbf{V}\backslash(\mathbf{J} \cup \mathbf{K})$, so we will not omit any homologous surrogate of any latent root variable. However, this is not a sufficient condition, an example is shown as Fig. 2, so this theorem can only be used to locate candidate homologous surrogate.

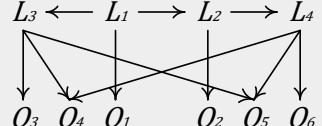

Figure 2: When $\mathbf{J} = \{L_1, L_2, L_3\}$ and $\mathbf{K} = \{O_1, O_2, O_3\}$, $O_4$ satisfies Thm. 2 but $O_4 \notin \text{HSu}(L_4)$.

**Proposition 1.** *Suppose $O_i$ and $O_j$ satisfy Thm. 2, then $\text{Pa}_\mathbf{L}(O_i)\backslash\mathbf{J} = \text{Pa}_\mathbf{L}(O_j)\backslash\mathbf{J}$ if and only if $\text{Cov}(\tilde{O}_i, O_j) \neq 0$.*

**Proposition 2.** *Suppose $O_i$ satisfies Thm. 2, then $O_i \in \text{HSu}(\text{Pa}_\mathbf{L}(O_i)\backslash\mathbf{J})$ if and only if $\forall O_j$ satisfying Thm. 2 and $\text{Pa}_\mathbf{L}(O_j)\backslash\mathbf{J} = \text{Pa}_\mathbf{L}(O_i)\backslash\mathbf{J}$, $\|\mathbf{M}_{\{O_i\}}^\mathbf{J}\|_0 \leq \|\mathbf{M}_{\{O_j\}}^\mathbf{J}\|_0$.*

**§ Estimating the Effects of Latent Root Variables.** This can be accomplished based on Cor. 3.

**Definition 3.** *(Cumulant) Given $n$ random variables $V_1, ..., V_n$, the $k$-th order cumulant is defined as a tensor of size $n \times ... \times n$ ($k$ times), whole element at position $(i_1, ..., i_k)$ is*

$$\text{Cum}(V_{i_1}, ..., V_{i_k}) = \sum_\pi (-1)^{|\pi|-1}(|\pi| - 1)! \prod_{B \in \pi} \mathbb{E}\left[\prod_{j \in B} V_j\right], \tag{7}$$

*where $\pi$ is enumerated over all partitions of $\{i_1, ..., i_k\}$.*

---

**Algorithm 1:** Partial recovery of the causal graph under Asmp. 1.

---

**Input: O**
**Output:** An($V$) for each $V \in \mathbf{V}$, $\mathbf{M_O^L}$, $\mathbf{M_O^O}$

1   Initialize $\mathbf{J} = \mathbf{K} = \emptyset$, and $\tilde{\mathbf{O}} = \mathbf{O}$.
2   **while** $\mathbf{K} \neq \mathbf{O}$ **do**
3     **while** *there exists* $O \in \mathbf{O}\backslash\mathbf{K}$ *satisfying* Thm. 1 **do**
4       Identify all observed root variables $\mathbf{O}'$ based on Thm. 1.
5       Estimate $\mathbf{M}_{\mathbf{O}\backslash\mathbf{K}}^{\mathbf{O}'}$ based on Cor. 1 and find De($O'$) for each $O' \in \mathbf{O}'$.
6       Remove $\mathbf{O}'$ based on Cor. 2.
7     **end**
8     Identify all latent root variables $\mathbf{L}'$ based on Thm. 2 plus Props. 1 and 2.
9     Estimate $\mathbf{M}_{\mathbf{O}\backslash\mathbf{K}}^{\mathbf{L}'}$ based on Cor. 3 and find An($L'$), De$_\mathbf{O}$($L'$) for each $L' \in \mathbf{L}'$.
10    Remove $\mathbf{L}'$ based on Cor. 4.
11 **end**

---

**Remark.** High-order cumulants have been widely used in the community of signal processing since the last century, especially in the topic of independent component analysis (ICA) (Thi & Jutten, 1995; Hyvärinen & Oja, 1997; Belkin et al., 2013; Voss et al., 2013; Ge & Zou, 2016), which is closely related to causal discovery (Shimizu et al., 2006; Hoyer et al., 2008).

**Corollary 3.** *Suppose $O_i$ satisfies Thm. 2 and $O_i \in \text{HSu}(L_i)$, then $\forall O_j \in \mathbf{O}\backslash(\mathbf{K} \cup \{O_i\})$,*

$$m_{O_i}^{L_i} m_{O_j}^{L_i} = \text{Cov}(\tilde{O}_i, O_j), \quad \left(\frac{m_{O_i}^{L_i}}{m_{O_j}^{L_i}}\right)^2 = \frac{\text{Cum}(\tilde{O}_i, \tilde{O}_i, \tilde{O}_i, O_j)}{\text{Cum}(\tilde{O}_i, O_j, O_j, O_j)}. \tag{8}$$

**Remark.** Let $m_{O_i}^{L_i} > 0$ without loss of generality, we can obtain $m_{O_i}^{L_i}$ and $m_{O_j}^{L_i}$. With the faithfulness assumption, $O_j$ is a descendant of $L_i$ if and only if $m_{O_j}^{L_i} \neq 0$, $L_h$ is an ancestor of $L_i$ if and only if $m_{O_i}^{L_h} \neq 0$ since An($L_i$) = An$_\mathbf{L}$($O_i$)$\backslash\{L_i\}$.

**§ Removing Latent Root Variables.** This can be accomplished based on Cor. 4.

**Corollary 4.** *Suppose $O_i$ satisfies Thm. 2 and $O_i \in \text{HSu}(L_i)$, if we update $\mathbf{J}$ to $\mathbf{J} \cup \{L_i\}$, $\mathbf{K}$ to $\mathbf{K} \cup \{O_i\}$, and $\tilde{O}_j$ to $\tilde{O}_j - (m_{O_j}^{L_i}/m_{O_i}^{L_i})\tilde{O}_i$ for each $O_j \in \mathbf{O}\backslash\mathbf{K}$, Cond. 1 is still valid.*

**§ Summary.** The algorithm is summarized in Alg. 1 with $O(|\mathbf{O}|^4)$ complexity, of which the process is shown as Fig. 3 and detailed below.

(1) Initially, $\mathbf{J} = \mathbf{K} = \emptyset$, $\tilde{O}_i = O_i$.

(2) First iteration shown as Fig. 3(b): Alg. 1 identifies $O_1$ as an observed root among $\mathbf{V}\backslash(\mathbf{J} \cup \mathbf{K})$, calculates $m_{O_i}^{O_1}$, and updates $\mathbf{K} := \mathbf{K} \cup \{O_1\}$, $\tilde{O}_i := \tilde{O}_i - m_{O_i}^{O_1}\tilde{O}_1$. There is De($O_1$) = $\{O_2\}$ (because $m_{O_2}^{O_1} \neq 0$). Next, it identifies $L_1$ as a latent root among $\mathbf{V}\backslash(\mathbf{J} \cup \mathbf{K})$ with HSu($L_1$) = $\{O_2\}$, calculates $m_{O_i}^{L_1}$, and updates $\mathbf{J} := \mathbf{J} \cup \{L_1\}$, $\mathbf{K} := \mathbf{K} \cup \{O_2\}$, $\tilde{O}_i := \tilde{O}_i - (m_{O_i}^{L_1}/m_{O_2}^{L_1})\tilde{O}_2$. There is De$_\mathbf{O}$($L_1$) = $\{O_2, ..., O_{11}\}$ (because $m_{O_2}^{L_1} \neq 0, ..., m_{O_{11}}^{L_1} \neq 0$).

(3) Second iteration shown as Fig. 3(c): Alg. 1 identifies no observed root among $\mathbf{V}\backslash(\mathbf{J} \cup \mathbf{K})$. Next, it identifies $L_2$ as a latent root among $\mathbf{V}\backslash(\mathbf{J} \cup \mathbf{K})$ with HSu($L_2$) = $\{O_3\}$, calculates $m_{O_i}^{L_2}$, and updates $\mathbf{J} := \mathbf{J} \cup \{L_2\}$, $\mathbf{K} := \mathbf{K} \cup \{O_3\}$, $\tilde{O}_i := \tilde{O}_i - (m_{O_i}^{L_2}/m_{O_3}^{L_2})\tilde{O}_3$. There is De$_\mathbf{O}$($L_2$) = $\{O_3, ..., O_{11}\}$ and An($L_2$) = $\{L_1\}$ (because $O_3 \in \text{HSu}(L_2)$ and $L_1 \in \text{An}(O_3)$).

(4) The following iterations proceed similarly. Particularly, at the last iteration shown as Fig. 3(f), $O_5, O_8, O_{10}, O_{11}$ are added into $\mathbf{K}$ progressively: Alg. 1 first identifies $O_{10}$ as an observed root among $\mathbf{V}\backslash(\mathbf{J} \cup \mathbf{K})$ and updates $\mathbf{K} := \mathbf{K} \cup \{O_{10}\}$, then it identifies $O_8, O_{11}$ as observed roots among $\mathbf{V}\backslash(\mathbf{J} \cup \mathbf{K})$ and updates $\mathbf{K} := \mathbf{K} \cup \{O_8, O_{11}\}$. Finally, it identifies $O_5$ as an observed root among $\mathbf{V}\backslash(\mathbf{J} \cup \mathbf{K})$ and updates $\mathbf{K} := \mathbf{K} \cup \{O_5\}$.

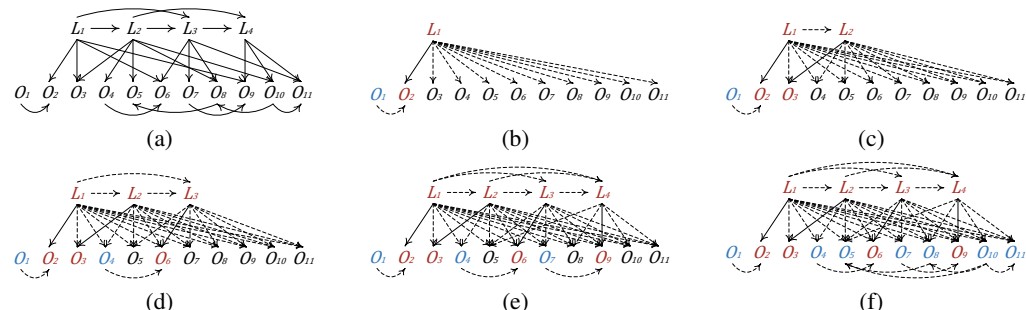

Figure 3: Illustration of Alg. 1, where solid arrow refers to parental relationship while dashed arrow refer to ancestral relationship. (a) Ground truth. (b-e) Result of each iteration.

**Theorem 3.** *Suppose the observed variables are generated by a LiNGAM with latent variables satisfying Asmp. 1, in the limit of infinite data, Alg. 1 identifies latent variables and ancestral relationships correctly.*

## 4    FULL IDENTIFICATION

**Definition 4.** *(Generalized homologous surrogate) $O \in \mathbf{O}$ is called a generalized homologous surrogate of $L \in \mathbf{L}$, denoted by $O \in \mathrm{GHSu}(L)$, if $O \in \mathrm{Ch}(L)$ and $\mathrm{Pa_L}(O) \subseteq \mathrm{An}(L) \cup \{L\}$.*

**Example.** In Fig. 1, $\mathrm{GHSu}(L_1) = \{O_2\}, \mathrm{GHSu}(L_2) = \{O_3, O_4, O_5\}, \mathrm{GHSu}(L_3) = \{O_6, O_7, O_8\}, \mathrm{GHSu}(L_4) = \{O_9, O_{10}, O_{11}\}$.

**Remark.** Trivially, if $O$ is $L$'s homologous surrogate, it must be $L$'s generalized homologous surrogate, but the reverse is not necessarily true. This is what "generalized" means.

**Intuition.** With causal relations between observed variables removed, for every latent variable, the ancestors of its each generalized homologous surrogate are exactly its ancestors plus itself. Because of this, generalized homologous surrogates can be located from observed variables.

**Assumption 2.** *Asmp. 1 holds and $\forall \{L_i, L_j\} \subseteq \mathbf{L}$ where $L_i \in \mathrm{An}(L_j)$, $\exists \{O_{j_1}, O_{j_2}\} \subseteq \mathrm{GHSu}(L_j)$ s.t $O_{j_1} \notin \mathrm{Ch}(L_i)$ and $O_{j_2} \notin \mathrm{Ch}(L_i)$.*

**Example.** The causal graph shown as Fig. 1 satisfies Asmp. 2.

**Remark.** An extremely special case where this assumption holds is that each latent variable has multiple observed pure children.

**Intuition.** Since Asmp. 1 holds, $\mathbf{M_O^L}$ and $\mathbf{M_O^O}$ can be estimated by Alg. 1. For every two latent variables $L_i, L_j$ where $L_i$ is an ancestor of $L_j$, $m_{L_j}^{Li}$ can be estimated through $L_j$'s two generalized homologous surrogates that are not $L_i$'s children, so we can also estimate $\mathbf{M_L^L}$. $\mathbf{A}$ can be readily recovered from $\mathbf{M}$ composed of $\mathbf{M_L^L}, \mathbf{0}, \mathbf{M_O^L}, \mathbf{M_O^O}$, that is, the causal graph can be fully recovered.

§ **High-level Overview.** Building on the partial recovery result, we first remove causal relations between observed variables. Second, with these relations removed, we locate generalized homologous surrogates of each latent variable (Lem. 1). Third, through these surrogates, we estimate $\mathbf{M_L^L}$ progressively (Thm. 4). Finally, given $\mathbf{M}$ composed of $\mathbf{M_L^L}, \mathbf{0}, \mathbf{M_O^L}, \mathbf{M_O^O}$, we recover $\mathbf{A}$.

§ **Removing Causal Relations Between Observed Variables.** $\mathbf{M_O^O}$ is estimated by Alg. 1, from which $\mathbf{A_O^O}$ can be recovered by $\mathbf{A_O^O} = \mathbf{I} - (\mathbf{M_O^O})^{-1}$ following Eq (5). Given $\mathbf{A_O^O}$, we can remove causal relations between observed variables. Specifically, for each $O_i \in \mathbf{O}$, we let

$$O_i^* = O_i - \sum_{O_j \in \mathrm{Pa}(O_i)} a_{O_i}^{O_j} O_j. \tag{9}$$

---

**Algorithm 2:** Full recovery of the causal graph under Asmp. 2.

---

**Input: O, L, $\mathbf{M_O^L}$, $\mathbf{M_O^O}$** returned by Alg. 1.
**Output: A**
1 Remove causal relations between **O** based on Eq. (9).
2 Locate generalized homologous surrogates of each latent variable based on Lem. 1.
3 $i := 1$
4 **while** *there exists $L \in \mathbf{L}$ s.t. $\mathrm{De}^i(L) \neq \emptyset$* **do**
5 $\quad$ Estimate $m_{L'}^L$, and $a_{O'}^L$ for each $L \in \mathbf{L}, L' \in \mathrm{De}^i(L), O' \in \mathrm{GHSu}(L')$ based on Thm. 4.
6 $\quad$ $i := i + 1$.
7 **end**
8 Recover **A** from **M** based on Eq. (5).

---

Note that $\mathbf{M_O^L}$ is also estimated by Alg. 1, then for each $L \in \mathbf{L}$,

$$m_{O_i^*}^L = m_{O_i}^L - \sum_{O_j \in \mathrm{Pa}(O_i)} a_{O_i}^{O_j} m_{O_j}^L. \tag{10}$$

**§ Locating Generalized Homologous Surrogates.** This can be accomplished based on Lem. 1.

**Lemma 1.** *$\forall L_i \in \mathbf{L}$ and $O_i \in \mathbf{O}$, $O_i \in \mathrm{GHSu}(L_i)$ if and only if $m_{O_i^*}^{L_i} \neq 0$ and $\forall O_j \in \mathbf{O}$ where $m_{O_j^*}^{L_i} \neq 0$, $\|\mathbf{M}_{\{O_i^*\}}^{\mathbf{L}}\|_0 \leq \|\mathbf{M}_{\{O_j^*\}}^{\mathbf{L}}\|_0$. Besides, there is $a_{O_i}^{L_i} = m_{O_i^*}^{L_i}$.*

**§ Estimating $\mathbf{M_L^L}$.** For ease of exposition, we introduce the concept of *n*-hop descendants (Def. 5). $\mathbf{M_L^L}$ is estimated in a progressive manner. Specifically, for each latent variable, we first estimate the mixing coefficient from itself to its every 1-hop descendant (Thm. 4(1)) and then the direct causal strength from itself to its every 1-hop descendant's each generalized homologous surrogate (Thm. 4(2)). On this basis, we investigate each latent variable's 2-hop descendants. Repeating this process, $\mathbf{M_L^L}$ can be estimated finally.

**Definition 5.** *(n-hop descendant) Given $\{L_i, L_j\} \subseteq \mathbf{L}$, we call $L_j$ is an n-hop descendant of $L_i$, denoted by $L_j \in \mathrm{De}^n(L_i)$, if $L_j \in \mathrm{De}(L_i)$ and the longest directed path from $L_i$ to $L_j$ has length n.*

**Remark.** Given the partial identification result, we can derive *n*-hop descendants for each latent variable and each *n*. Specifically, suppose $L_j \in \mathrm{De}(L_i)$, we find the longest array that starts with $L_i$ and ends with $L_j$, in which any variable is an ancestor of the variable that follows it. If the array has length $n + 1$, then $L_j \in \mathrm{De}^n(L_i)$.

**Theorem 4.** *Suppose $\{L_i, L_j\} \subseteq \mathbf{L}$, $L_j \in \mathrm{De}^n(L_i)$. $\forall O_j \in \mathrm{GHSu}(L_j)$, let*

$$\mu_{O_j^*}^{L_i} = m_{O_j^*}^{L_i} - \sum_{L_k \in \mathrm{De}(L_i) \cap \mathrm{An}(L_j)} m_{L_k}^{L_i} a_{O_j}^{L_k}. \tag{11}$$

(a) *There exists $\{O_{j_1}, O_{j_2}\} \subseteq \mathrm{GHSu}(L_j)$ s.t. $\mu_{O_{j_1}^*}^{L_i}/a_{O_{j_1}}^{L_j} = \mu_{O_{j_2}^*}^{L_i}/a_{O_{j_2}}^{L_j}$ and $m_{L_j}^{L_i} = \mu_{O_{j_1}^*}^{L_i}/a_{O_{j_1}}^{L_j}$.*

(b) *$a_{O_j}^{L_i} = \mu_{O_j^*}^{L_i} - m_{L_j}^{L_i} a_{O_j}^{L_j}$.*

**Remark.** Given $L_j \in \mathrm{De}^n(L_i)$ and $O_j \in \mathrm{GHSu}(L_j)$, if we have already investigated $\mathrm{De}^1(L), ..., \mathrm{De}^{n-1}(L)$ for each $L \in \mathbf{L}$, then $m_{L_k}^{L_i}$ and $a_{O_j}^{L_k}$ in RHS of Eq. (11) are known. Moreover, $m_{O_j^*}^{L_i}$ in RHS of Eq. (11) can be derived from Eq. (10), so $\mu_{O_j^*}^{L_i}$ in LHS of Eq. (11) is known. In fact, $\mu_{O_j^*}^{L_i}$ is exactly $a_{O_j}^{L_i} + m_{L_j}^{L_i} a_{O_j}^{L_j}$ (see Eq. (53) in App. C.3). With $\mu_{O_j^*}^{L_i}$ known, (a) provides a method to estimate $m_{L_j}^{L_i}$ through $L_j$'s some two generalized homologous surrogate $O_{j_1}$ and $O_{j_2}$, where $O_{j_1}$ and $O_{j_2}$ are exactly those variables that are not $L_i$'s children (see proof in App. C.3), and (b) provides a method to estimate $a_{O_j}^{L_i}$ for each $O_j \in \mathrm{GHSu}(L_j)$ using the just estimated $m_{L_j}^{L_i}$.

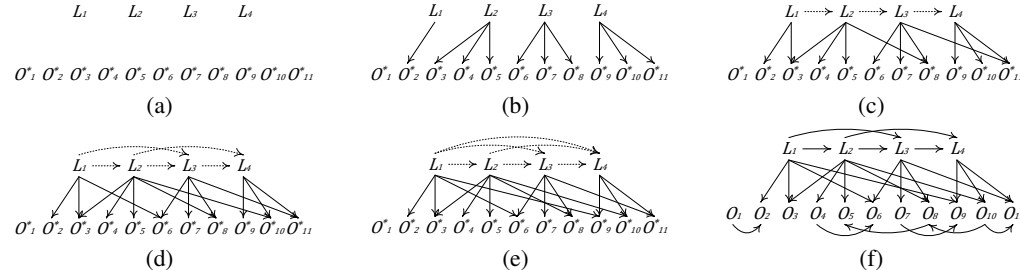

Figure 4: Illustration of Alg. 2, where solid arrow refers to parental relationship while dashed arrow refer to ancestral relationship. (a) Removing causal relations between observed variables. (b) Identifying generalized homologous surrogates of each $L \in \mathbf{L}$. (c-e) Investigation to $\mathrm{De}^1(L), \mathrm{De}^2(L), \mathrm{De}^3(L)$ for each $L \in \mathbf{L}$. (f) Recovering $\mathbf{A}$ from $\mathbf{M}$.

**Example.** In Fig. 1, $L_3 \in \mathrm{De}^2(L_1)$ and $\mathrm{GHSu}(L_3) = \{O_6, O_7, O_8\}$. According to Eq. (11), $\mu_{O_6^*}^{L_1} = m_{O_6^*}^{L_1} - m_{L_2}^{L_1} a_{O_6}^{L_2}$, $\mu_{O_7^*}^{L_1} = m_{O_7^*}^{L_1} - m_{L_2}^{L_1} a_{O_7}^{L_2}$, and $\mu_{O_8^*}^{L_1} = m_{O_8^*}^{L_1} - m_{L_2}^{L_1} a_{O_8}^{L_2}$. Then we can calculate $m_{L_3}^{L_1}$ based on (a) and $a_{O_6}^{L_1}, a_{O_7}^{L_1}, a_{O_8}^{L_1}$ based on (b).

§ **Recovering A.** $\mathbf{M}$ is composed of $\mathbf{M_L^L}, \mathbf{0}, \mathbf{M_O^L}, \mathbf{M_O^O}$ where $\mathbf{M_L^L}$ is just estimated and $\mathbf{M_O^L}, \mathbf{M_O^O}$ are estimated by Alg. 1, so we can readily recover $\mathbf{A}$ from $\mathbf{M}$ by $\mathbf{M} = (\mathbf{I} - \mathbf{A})^{-1}$ following Eq (5).

§ **Summary.** The algorithm is summarized in Alg. 2 with $O(|\mathbf{O}|^2 |\mathbf{L}|^2)$ complexity. With the result returned by Alg. 1 as the input, its procedures are shown as Fig. 4 and detailed below.

(1) Fig. 4(a): Alg. 2 removes causal relations between observed variables.

(2) Fig. 4(b): Alg. 2 locates generalized homologous surrogates. Specifically, it determines $\mathrm{GHSu}(L_1) = \{O_2\}$, $\mathrm{GHSu}(L_2) = \{O_3, O_4, O_5\}$, $\mathrm{GHSu}(L_3) = \{O_6, O_7, O_8\}$, $\mathrm{GHSu}(L_4) = \{O_9, O_{10}, O_{11}\}$. Also, it determines $\mathbf{A}_{\{O_2\}}^{\{L_1\}}, \mathbf{A}_{\{O_3, O_4, O_5\}}^{\{L_2\}}, \mathbf{A}_{\{O_6, O_7, O_8\}}^{\{L_3\}}, \mathbf{A}_{\{O_9, O_{10}, O_{11}\}}^{\{L_4\}}$.

(3) Fig. 4(c): Alg. 2 investigates $\mathrm{De}^1(L)$ for each $L \in \mathbf{L}$: $\mathrm{De}^1(L_1) = \{L_2\}$, $\mathrm{De}^1(L_2) = \{L_3\}$, $\mathrm{De}^1(L_3) = \{L_4\}$. It estimates $m_{L_2}^{L_1}, m_{L_3}^{L_2}, m_{L_4}^{L_3}$ and then $\mathbf{A}_{\{O_3, O_4, O_5\}}^{\{L_1\}}, \mathbf{A}_{\{O_6, O_7, O_8\}}^{\{L_2\}}, \mathbf{A}_{\{O_9, O_{10}, O_{11}\}}^{\{L_3\}}$ where only $a_{O_3}^{L_1}, a_{O_8}^{L_2}, a_{O_{11}}^{L_3}$ are non-zero.

(4) Fig. 4(d): Alg. 2 investigates $\mathrm{De}^2(L)$ for each $L \in \mathbf{L}$: $\mathrm{De}^2(L_1) = \{L_3\}$, $\mathrm{De}^2(L_2) = \{L_4\}$. It estimates $m_{L_3}^{L_1}, m_{L_4}^{L_2}$ and then $\mathbf{A}_{\{O_6, O_7, O_8\}}^{\{L_1\}}, \mathbf{A}_{\{O_9, O_{10}, O_{11}\}}^{\{L_2\}}$ where only $a_{O_6}^{L_1}, a_{O_{10}}^{L_2}$ are non-zero.

(5) Fig. 4(d): Alg. 2 investigates $\mathrm{De}^3(L)$ for each $L \in \mathbf{L}$: $\mathrm{De}^3(L_1) = \{L_4\}$. It estimates $m_{L_4}^{L_1}$ and then $\mathbf{A}_{\{O_9, O_{10}, O_{11}\}}^{\{L_1\}}$ where only $a_{O_9}^{L_1}$ is non-zero.

(6) Fig. 4(f): Alg. 2 recovers $\mathbf{A}$ from $\mathbf{M}$.

**Theorem 5.** *Suppose the observed variables are generated by a LiNGAM with latent variables satisfying Asmp. 2, in the limit of infinite data, Algs. 1 and 2 together identifies latent variables and parental relationships correctly.*

## 5 EXPERIMENT

We first use four causal graphs shown as Fig. 5 to generate synthetic data. For each causal graph, we draw 10 sample sets of size 5k, 10k, 20k respectively. Each direct causal strength is sampled from a uniform distribution over $[-2.0, -0.5] \cup [0.5, 2.0]$ and each exogenous noise is generated from exponential distribution. We compare our methods with GIN (Xie et al., 2020), LaHME (Xie et al., 2022), and PO-LiNGAM (Jin et al., 2024). We use 3 metrics to evaluate their performances: (1) *Error in Latent Variables*, the absolute difference between the estimated number of latent variables and the ground-truth one; (2) *Correct-Ordering Rate*, the number of correctly estimated causal orderings divided by that of ground-truth causal orderings; (3) *F1-Score* of causal edges. Results are summarized in Tab. 1, where we also report the running time. In particular, we set the size of the largest atomic unit in GIN and PO-LiNGAM to 1 for a fair comparison.

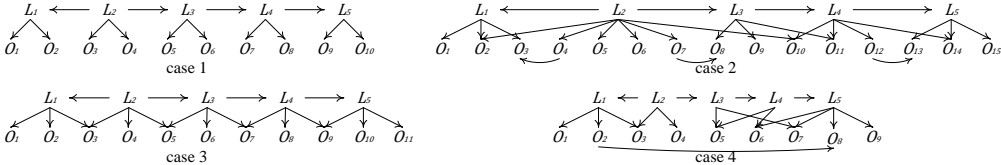

Figure 5: Causal graphs used to generate synthetic data.

Table 1: Comparison on synthetic data. ↑ means higher is better while ↓ means lower is better.

| | | Error in Latent Variables ↓ | | | Correct-Ordering Rate ↑ | | | F1-Score ↑ | | | Running Time(s) ↓ | | |
|---|---|---|---|---|---|---|---|---|---|---|---|---|---|
| | | 5k | 10k | 20k | 5k | 10k | 20k | 5k | 10k | 20k | 5k | 10k | 20k |
| Case 1 | GIN | **0.0±0.0** | 0.2±0.4 | **0.0±0.0** | **1.00±0.00** | 0.95±0.11 | **1.00±0.00** | **1.00±0.00** | 0.97±0.07 | **1.00±0.00** | **1.56±0.13** | **1.74±0.14** | **2.07±0.18** |
| | LaHME | 1.1±0.3 | 1.3±0.6 | 1.1±0.3 | 0.87±0.18 | 0.78±0.21 | 0.84±0.15 | 0.79±0.04 | 0.76±0.06 | 0.76±0.07 | 2.01±0.14 | 2.32±0.25 | 2.96±0.23 |
| | PO-LiNGAM | 0.8±0.6 | 0.5±0.5 | 0.1±0.3 | 0.87±0.09 | 0.91±0.10 | 0.98±0.05 | 0.77±0.16 | 0.84±0.14 | 0.98±0.06 | 121.62±27.26 | 116.31±27.64 | 117.39±16.61 |
| | Ours | 0.4±0.5 | **0.1±0.3** | **0.0±0.0** | 0.94±0.07 | **0.99±0.03** | **1.00±0.00** | 0.92±0.10 | **0.98±0.05** | 0.99±0.02 | 2.64±0.27 | 3.05±0.27 | 4.01±0.08 |
| Case 2 | GIN | 3.8±0.4 | 3.9±0.3 | 4.1±0.5 | 0.07±0.04 | 0.06±0.03 | 0.05±0.04 | 0.18±0.07 | 0.15±0.05 | 0.13±0.08 | **3.13±0.31** | **3.47±0.33** | **4.13±0.44** |
| | LaHME | 1.6±1.0 | 1.7±0.9 | 2.1±1.0 | 0.43±0.09 | 0.45±0.04 | 0.33±0.12 | 0.43±0.06 | 0.41±0.06 | 0.39±0.07 | 36.60±14.43 | 86.45±71.08 | 116.15±96.61 |
| | PO-LiNGAM | 3.9±1.3 | 4.3±1.2 | 3.9±1.4 | 0.20±0.27 | 0.12±0.25 | 0.12±0.20 | 0.15±0.18 | 0.12±0.15 | 0.17±0.23 | 2214.19±779.92 | 2073.97±678.85 | 2482.57±704.83 |
| | Ours | **1.2±0.6** | **0.7±0.8** | **0.2±0.4** | **0.82±0.09** | **0.91±0.10** | **0.97±0.05** | **0.77±0.11** | **0.88±0.13** | **0.94±0.07** | 6.27±0.54 | 7.72±0.33 | 9.69±0.72 |
| Case 3 | GIN | 3.0±0.4 | 3.0±0.0 | 3.0±0.0 | 0.12±0.05 | 0.11±0.00 | 0.11±0.00 | 0.33±0.06 | 0.33±0.00 | 0.33±0.00 | **1.64±0.12** | **1.79±0.11** | **2.22±0.11** |
| | LaHME | 2.0±1.0 | 2.5±1.0 | 2.7±0.9 | 0.37±0.24 | 0.24±0.17 | 0.19±0.09 | 0.58±0.15 | 0.48±0.10 | 0.46±0.07 | 6.36±1.18 | 8.64±1.13 | 10.26±1.19 |
| | PO-LiNGAM | 1.5±0.7 | **0.8±0.9** | 0.6±0.9 | 0.53±0.19 | 0.56±0.17 | 0.60±0.17 | 0.48±0.13 | 0.52±0.08 | 0.54±0.07 | 295.43±79.55 | 289.59±40.06 | 369.12±31.68 |
| | Ours | **1.4±0.8** | **0.8±0.4** | **0.5±0.5** | **0.71±0.15** | **0.84±0.07** | **0.91±0.08** | **0.63±0.17** | **0.76±0.06** | **0.81±0.07** | 2.58±0.34 | 3.23±0.28 | 4.36±0.38 |
| Case 4 | GIN | 4.8±0.4 | 4.9±0.3 | 5.0±0.0 | 0.01±0.02 | 0.01±0.02 | 0.00±0.00 | 0.04±0.08 | 0.02±0.06 | 0.00±0.00 | **0.79±0.07** | **0.87±0.08** | **1.05±0.08** |
| | LaHME | 4.0±0.0 | 4.2±0.4 | 4.1±0.03 | 0.11±0.00 | 0.09±0.05 | 0.10±0.03 | 0.30±0.00 | 0.24±0.12 | 0.27±0.09 | 11.88±1.48 | 13.61±1.67 | 17.51±0.95 |
| | PO-LiNGAM | 4.9±0.3 | 4.8±0.4 | 4.9±0.3 | 0.01±0.03 | 0.02±0.04 | 0.01±0.02 | 0.03±0.01 | 0.04±0.08 | 0.02±0.06 | 63.39±16.79 | 69.07±17.81 | 89.61±26.58 |
| | Ours | **2.1±1.1** | **2.1±1.1** | 3.0±0.0 | **0.44±0.21** | **0.41±0.18** | **0.25±0.01** | **0.53±0.16** | **0.53±0.09** | **0.50±0.02** | 1.68±0.28 | 1.85±0.11 | 2.30±0.08 |

In case 1, each latent variable has at least two observed pure children, GIN and our algorithm demonstrate optimal performance. In other cases, the pure children assumption is not valid, so previous methods cannot handle these cases properly. In case 2, Asmp. 2 is valid, so our algorithm significantly outperforms others. In case 3, although Asmp. 2 is invalid, Asmp. 1 holds, so our algorithm still reaches the best performance, especially a high correct-ordering rate. In case 4, the violation of Asmp. 1 leads to a remarkable degradation in the performance of our algorithm, although it still exhibits a remarkable advantage over others since it can still identify $L_1$ and $L_2$. Moreover, in most cases, our algorithm is far more efficient than both LaHME and PO-LiNGAM. This is because LaHME has factorial complexity w.r.t. the number of variables in the worse case while PO-LiNGAM has exponential time complexity in the worst case.

Although our algorithm eliminates the need for pure children, we acknowledge that it cannot yield satisfactory results when the sample size is small. For instance, in Case 2 where Asmp. 2 holds, our algorithm can achieve 0 error in latent variables, 1 correct-ordering rate, and 1 F1-score theoretically, but it performs poorly in practice when the sample size is 5k. This can be attributed to two main factors. First, our algorithm estimates the mixing coefficients from latent to observed variables through high-order cumulants. Compared to covariances, high-order cumulants are more sensitive to extreme values and outliers, especially when the sample size is small. Second, our algorithm operates in a progressive manner, of which each step builds upon the previous one, so errors are propagated and amplified during this process.

Besides synthetic data, we also evaluate our algorithm on a real-world dataset Holzinger and Swineford 1939 (Rosseel, 2012). More details are provided in App. D.

# 6 CONCLUSION

In this paper, we introduce a new concept called homologous surrogate to facilitate causal discovery with latent variables, which fundamentally differ from the pure child. We formulate two assumptions involving homologous surrogates and develop a series of novel theoretical results under each assumption, implying that the causal graph can be partially/fully recovered under the weaker/stronger assumption. Also, building on our theoretical results, we derive an algorithm that fully utilizes the properties homologous surrogates for causal graph recovery. Our work broadens the applicability of causal discovery and may benefit research in natural and social sciences.

**Limitations.** First, our algorithm cannot handle the latent hierarchical structure where some latent variables have no observed children. Second, this work does not accommodate non-stationary (Liu & Kuang, 2023) and cyclic (Sethuraman et al., 2023) causal relations. We will endeavor to overcome these limitations in the future.

## ACKNOWLEDGMENTS

Tongliang Liu is partially supported by the following Australian Research Council projects: FT220100318, DP220102121, LP220100527, LP220200949, IC190100031. Xiu-Chuan Li is partially supported by ARC FT220100318. Jun Wang is partially supported by the JD Technology Scholarship for Postgraduate Research in Artificial Intelligence.

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

# A    RELATED WORKS

Most traditional causal discovery methods (Spirtes & Glymour, 1991; Colombo et al., 2014; Le et al., 2016; Chickering, 2002; Shimizu et al., 2006; Hoyer et al., 2009; Zhang & Hyvärinen, 2009; Peters et al., 2014; Mooij et al., 2016) assume the absence of latent variables. Since latent variables are ubiquitous in real-world scenarios, extensive research has been devoted to causal discovery with latent variables. Our work is most related to those that not only allow causally-related latent variables but also can identify latent variables along with their causal relations, most of which make the pure children assumption. More specifically, some works (Silva et al., 2006; Shimizu et al., 2009; Kummerfeld & Ramsey, 2016; Cai et al., 2019; Chen et al., 2022; Zeng et al., 2021; Xie et al., 2022; Chen et al., 2023) make the special pure children assumption that each latent variable has multiple pure children, where a variable is said a pure child of anther only if the latter is the only parent of the former. Others (Xie et al., 2020; 2024; Huang et al., 2022; Dong et al., 2024; Jin et al., 2024) make the general pure children assumption that each latent variable belongs to a latent set (comprising one or more latent variables) which has sufficient pure children, where a variable is said a pure child of a latent set only if all parents of the former are within the latter. Although the general pure children assumption is weaker than the special one, it comes at the cost of local unidentifiability. Specifically, if a latent set contains multiple latent variables, none of which has its own pure children, even the existence of causal relations between these latent variables cannot be determined, let alone their directions. By introducing the concept of homologous surrogates, our work eliminates the need for pure children, in stark contrast to the above studies.

Although the pure children assumption has been widely adopted by previous works, Adams et al. (2021) argue that it is restrictive in practice and also not necessary for identifiability of linear latent non-Gaussian models in theory. They develop a causal discovery algorithm under the assumption which is exactly sufficient and necessary for identifiability. Unfortunately, this algorithm is unpractical as acknowledged by themselves. First, it estimates $\mathbf{M_O^V}$ via overcomplete independent components analysis, which requires the number of latent variables as prior knowledge and is computationally intractable. Second, to recover $\mathbf{A}$ from $\mathbf{M_O^V}$, it needs to test which submatrices' singular values are exact zeros, which is quite sensitive to noises. Recently, Li et al. (2024) suggest that a pseudo-pure pair, composed of two adjacent observed variables that both become pure children of a same latent variable if the edge between them is removed, can be transformed into two pure children under certain assumption. Based on this finding, they propose a practical causal discovery algorithm free from the pure children assumption. Clearly, our work diverges significantly from existing works these two studies, offering a novel perspective on causal discovery with latent variables.

While the works discussed above all focus on the linear case, several studies have ventured into nonlinear problems, but most assume access to counterfactual data (Brehmer et al., 2022; Ahuja et al., 2022) or interventional data (Ahuja et al., 2023; Jiang & Aragam, 2023; Buchholz et al., 2023; Zhang et al., 2023). Notably, without structural restrictions such as our homologous surrogates assumption, even linear causal models are unidentifiable without comprehensive interventional data obtained by intervening on each latent variable individually (Squires et al., 2023). To the best of our knowledge, only Kivva et al. (2021) and Kong et al. (2023) can handle non-linear problems with latent variables using solely observational data, but they both make strong assumptions, e.g., all latent variables are discrete and the mapping from all exogenous noises to observed variables are invertible. We leave further research on nonlinear problems to our future work.

# B    NOTATIONS

We summarize notations in Tab. 2

Table 2: Summary of notations.

| Notation | Description | First appeared |
|---|---|---|
| $\mathcal{G}$ | Causal graph | Sec. 2 |
| $\mathbf{L}$ | Set of latent variables | Sec. 2 |
| $\mathbf{O}$ | Set of observed variables | Sec. 2 |
| $\mathbf{V}$ | $\mathbf{L} \cup \mathbf{O}$ | Sec. 2 |
| $L_i$ | A latent variable | Sec. 2 |
| $O_i$ | An observed variable | Sec. 2 |
| $V_i$ | A latent or observed variable | Sec. 2 |
| $\epsilon_{V_i}$ | Exogenous noise of $V_i$ | Sec. 2 |
| $\text{Pa}(V)$ | Parents of $V$ | Sec. 2 |
| $\text{Pa}(\mathbf{V}')$ | $\bigcup_{V \in \mathbf{V}'} \text{Pa}(V)$ | Sec. 2 |
| $\text{Pa}_{\mathbf{L}}(V)$ | Latent parents of $V$ | Sec. 2 |
| $\text{Pa}_{\mathbf{O}}(V)$ | Observed parents of $V$ | Sec. 2 |
| $\text{Ch}(V)$ | Children of $V$ | Sec. 2 |
| $\text{Ne}(V)$ | Neighbors of $V$ | Sec. 2 |
| $\text{An}(V)$ | Ancestors of $V$ | Sec. 2 |
| $\text{De}(V)$ | Descendants of $V$ | Sec. 2 |
| $a_{V_j}^{V_i}$ | direct causal strength from $V_i$ to $V_j$ | Sec. 2 |
| $m_{V_j}^{V_i}$ | mixing coefficient from $V_i$ to $V_j$ | Sec. 2 |
| $\mathbf{A}_{\mathbf{V}_2}^{\mathbf{V}_1}$ | adjacency matrix from $\mathbf{V}_1$ to $\mathbf{V}_2$ | Sec. 2 |
| $\mathbf{M}_{\mathbf{V}_2}^{\mathbf{V}_1}$ | mixing matrix from $\mathbf{V}_1$ to $\mathbf{V}_2$ | Sec. 2 |
| $\text{HSu}(V)$ | Homologous surrogates of $V$ | Def. 1 in Sec. 3 |
| $\mathbf{J}$ | Set of removed latent variables | Sec. 3 |
| $\mathbf{K}$ | Set of removed observed variables | Sec. 3 |
| $\tilde{O}_i$ | Auxiliary variable of $O_i$ satisfying Cond. 1(2) | Sec. 3 |
| $\text{R}(V_1, V_2|V_3)$ | Pseudo-residual of $V_1, V_2$ relative to $V_3$ | Def. 2 in Sec. 3 |
| $\text{GHSu}(V)$ | Generalized homologous surrogates of $V$ | Def. 4 in Sec. 4 |
| $O_i^*$ | $O_i$ with its all observed parents removed | Eq (9) in Sec. 4 |
| $\text{De}^n(L)$ | $n$-hop descendants of $L$ | Def. 5 in Sec. 4 |
| $\mathbf{K}_1$ | $\{O \in \mathbf{K}|\text{Ch}(O) \neq \emptyset\}$ | Eq. (15) in App. C.2 |
| $\mathbf{K}_2$ | $\{O \in \mathbf{K}|\text{Ch}(O) = \emptyset\}$ | Eq. (16) in App. C.2 |
| $\mathcal{G}^*$ | $\mathcal{G}$ with all causal relations between observed variables removed | App. C.3 |
| $\mathbf{O}^*$ | $\{O^*|O \in \mathbf{O}\}$ | App. C.3 |

## C PROOF

### C.1 IMPORTANT LEMMAS

**Darmois-Skitovitch (D-S) Theorem.** (Kagan et al., 1973) Suppose two random variables $V_1$ and $V_2$ are both linear combinations of independent random variables $\{n_i\}_i$:

$$V_1 = \sum_i \alpha_i n_i, \quad V_2 = \sum_i \beta_i n_i. \tag{12}$$

Then, if $V_1 \perp\!\!\!\perp V_2$, each $n_i$ for which $\alpha_i \beta_i \neq 0$ follows Gaussian distribution. That is, if there exists a non-Gaussian $n_j$ s.t. $\alpha_j \beta_j \neq 0$, $V_1 \not\perp\!\!\!\perp V_2$.

**Lemma 2.** *If three variables $V_1, V_2, V_3$ can be expressed as*

$$V_1 = \gamma_1 e + e_1, \quad V_2 = \gamma_2 e + e_2, \quad V_3 = \gamma_3 e + e_3, \tag{13}$$

*where $e, (e_1, e_2), e_3$ are mutually independent and $\gamma_1 \gamma_2 \gamma_3 \neq 0$, then $\mathrm{R}(V_1, V_2|V_3) \perp\!\!\!\perp V_3$.*

*Proof.*

$$\mathrm{R}(V_1, V_2|V_3) = (\gamma_1 e + e_1) - \frac{\mathrm{Cov}(\gamma_1 e + e_1, \gamma_3 e + e_3)}{\mathrm{Cov}(\gamma_2 e + e_2, \gamma_3 e + e_3)}(\gamma_2 e + e_2) = e_1 - \frac{\gamma_1}{\gamma_2} e_2 \perp\!\!\!\perp V_3. \tag{14}$$

$\square$

**Lemma 3.** *Given three variables $V_1, V_2, V_3$ where $\mathrm{Cov}(V_1, V_3)\mathrm{Cov}(V_2, V_3) \neq 0$, if $\exists V \in \mathbf{V}$ s.t. only one of $m_{V_1}^V$ and $m_{V_2}^V$ is non-zero and $m_{V_3}^V$ is non-zero, then $\mathrm{R}(V_1, V_2|V_3) \not\perp\!\!\!\perp V_3$.*

*Proof.* Since $\mathrm{Cov}(V_1, V_3)\mathrm{Cov}(V_2, V_3) \neq 0$ and only one of $m_{V_1}^V$ and $m_{V_2}^V$ is non-zero, $\mathrm{R}(V_1, V_2|V_3)$ contains $\epsilon_V$. Because $m_{V_3}^V \neq 0$, $\mathrm{R}(V_1, V_2|V_3) \not\perp\!\!\!\perp V_3$ based on D-S Theorem. $\square$

## C.2 Proof of Theoretical Results in Sec. 3

**Definition 1.** (homologous surrogate) $O \in \mathbf{O}$ is called a homologous surrogate of $L \in \mathbf{L}$, denoted by $O \in \mathrm{HSu}(L)$, if $O \in \mathrm{Ch}(L)$, $\mathrm{Ch}(O) = \emptyset$, $\mathrm{An}_{\mathbf{L}}(O) = \mathrm{An}(L) \cup \{L\}$ and $\mathrm{An}_{\mathbf{O}}(O) \cap \mathrm{De}_{\mathbf{O}}(L) = \emptyset$.

**Assumption 1.** $\forall L \in \mathbf{L}$, $\mathrm{HSu}(L) \neq \emptyset$ and $|\mathrm{Ch}(L)| \geq 2$.

**Remark.** We can easily derive from Asmp. 1 that $\forall L \in \mathbf{L}$, $|\mathrm{De}_{\mathbf{O}}(L)| \geq 2$.

**Condition 1.** (1) For each $V \in \mathbf{V}\backslash(\mathbf{J} \cup \mathbf{K})$, $\mathrm{De}(V) \cap (\mathbf{J} \cup \mathbf{K}) = \emptyset$. (2) For each $L \in \mathbf{J}$ and $O \in \mathbf{K}$ where $\mathrm{Ch}(O) \neq \emptyset$, $m_{\tilde{O}_i}^L = m_{\tilde{O}_i}^O = 0$.

**Remark.** Let $\mathbf{K}_1 = \{O \in \mathbf{K}|\mathrm{Ch}(O) \neq \emptyset\}$ and $\mathbf{K}_2 = \{O \in \mathbf{K}|\mathrm{Ch}(O) = \emptyset\}$. Based on Cond. 1, each $O_i \in \mathbf{O}\backslash\mathbf{K}$ and corresponding $\tilde{O}_i$ (which, as mentioned in the main text, is a linear combination of $O_i$ and variables in $\mathbf{K}$ where the coefficient of $O_i$ is always 1 while that of each variable in $\mathbf{K}$ is not fixed) can be expressed as

$$O_i = \sum_{L_i \in \mathbf{J}} m_{O_i}^{L_i} \epsilon_{L_i} + \sum_{L_j \in \mathbf{L}\backslash\mathbf{J}} m_{O_i}^{L_j} \epsilon_{L_j} + \sum_{O_{j_1} \in \mathbf{K}_1} m_{O_i}^{O_{j_1}} \epsilon_{O_{j_1}} + \sum_{O_k \in \mathbf{O}\backslash\mathbf{K}} m_{O_i}^{O_k} \epsilon_{O_k}, \tag{15}$$

$$\tilde{O}_i = \sum_{L_j \in \mathbf{L}\backslash\mathbf{J}} m_{O_i}^{L_j} \epsilon_{L_j} + \sum_{O_{j_2} \in \mathbf{K}_2} \lambda_{ij_2} \epsilon_{O_{j_2}} + \sum_{O_k \in \mathbf{O}\backslash\mathbf{K}} m_{O_i}^{O_k} \epsilon_{O_k}. \tag{16}$$

**Lemma 4.** *If $\exists V_i \in \mathbf{V}\backslash(\mathbf{J} \cup \mathbf{K})$ and $\{O_i, O_j\} \subseteq \mathbf{O}\backslash\mathbf{K}$ s.t. $m_{O_i}^{V_i} m_{O_j}^{V_i} \neq 0$, then $\mathrm{Cov}(O_i, \tilde{O}_j) \neq 0$.*

*Proof.* Based on Eqs. (15) and (16),

$$\mathrm{Cov}(O_i, \tilde{O}_j) = \sum_{V \in \mathbf{V}\backslash(\mathbf{J} \cup \mathbf{K})} m_{O_i}^V m_{O_j}^V \mathrm{Var}(\epsilon_V). \tag{17}$$

According to the faithfulness assumption, $\mathrm{Cov}(O_i, \tilde{O}_j) \neq 0$. $\square$

**Theorem 1.** *Suppose $O_i \in \mathbf{O}\backslash\mathbf{K}$, then $\mathrm{An}(O_i) \subseteq (\mathbf{J} \cup \mathbf{K})$ if and only if $\forall O_j \in \mathbf{O}\backslash(\mathbf{K} \cup \{O_i\})$, $\mathrm{R}(O_j, O_i|\tilde{O}_i) \perp\!\!\!\perp \tilde{O}_i$.*

**Proof sketch.** If $\mathrm{An}(O_i) \subseteq (\mathbf{J} \cup \mathbf{K})$, we can prove independence for $O_j \notin \mathrm{De}(O_i)$ easily (in this case, $\mathrm{R}(O_j, O_i|\tilde{O}_i) = O_j \perp\!\!\!\perp \tilde{O}_i$) and prove independence for $O_j \in \mathrm{De}(O_i)$ based on Lem. 2 ($\epsilon_{O_i}$ serves as $e$ in Eq. (13)). Otherwise, there exists an $O_j \in \mathbf{O}\backslash(\mathbf{K} \cup \{O_i\})$ s.t. $\mathrm{Cov}(\tilde{O}_i, O_j) \neq 0$ and $m_{O_j}^{O_i} = 0$, so dependence can be proven by Lem. 3.

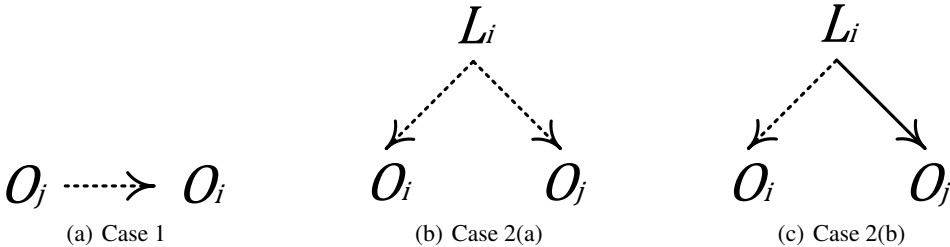

(a) Case 1          (b) Case 2(a)          (c) Case 2(b)

Figure 6: Illustration of "If" part in proof of Thm. 1. A dotted arrow from $V_1$ to $V_2$ means that $V_2 \in \mathrm{De}(V_1)$.

*Proof.* "Only if": Based on Eqs. (15) and (16),

$$O_j = \sum_{L_i \in \mathbf{J}} m^{L_i}_{O_j} \epsilon_{L_i} + \sum_{L_j \in \mathbf{L} \backslash \mathbf{J}} m^{L_j}_{O_j} \epsilon_{L_j} + \sum_{O_{k_1} \in \mathbf{K}_1} m^{O_{k_1}}_{O_j} \epsilon_{O_{k_1}} + \sum_{O_l \in \mathbf{O} \backslash \mathbf{K}} m^{O_l}_{O_j} \epsilon_{O_l}, \tag{18}$$

$$O_i = \sum_{L_i \in \mathbf{J}} m^{L_i}_{O_i} \epsilon_{L_i} + \sum_{O_{k_1} \in \mathbf{K}_1} m^{O_{k_1}}_{O_i} \epsilon_{O_{k_1}} + \epsilon_{O_i}, \tag{19}$$

$$\tilde{O}_i = \sum_{O_{k_2} \in \mathbf{K}_2} \lambda_{ik_2} \epsilon_{O_{k_2}} + \epsilon_{O_i}. \tag{20}$$

If $m^{O_i}_{O_j} = 0$, then $\mathrm{Cov}(\tilde{O}_i, O_j) = 0$, $\mathrm{R}(O_j, O_i | \tilde{O}_i) = O_j \perp\!\!\!\perp \tilde{O}_i$. Otherwise, based on Lem. 2, there is also $\mathrm{R}(O_j, O_i | \tilde{O}_i) \perp\!\!\!\perp \tilde{O}_i$, where $\epsilon_{O_i}$ serves as $e$ in Eq. (13).

"If": We prove this part by contradiction. Suppose $\mathrm{An}(O_i) \nsubseteq (\mathbf{J} \cup \mathbf{K})$. There are two possible cases as follows.

1. Suppose $\mathrm{An}_{\mathbf{O}}(O_i) \nsubseteq \mathbf{K}$, let $O_j \in (\mathbf{O} \backslash \mathbf{K}) \cap \mathrm{An}_{\mathbf{O}}(O_i)$. An illustrative example is shown as Fig. 6(a). As $m^{O_j}_{O_i} m^{O_j}_{O_j} \neq 0$, $\mathrm{Cov}(O_j, \tilde{O}_i) \neq 0$ based on Lem. 4. Also, it is trivial that $\mathrm{Cov}(O_i, \tilde{O}_i) \neq 0$. Since $m^{O_i}_{O_i} m^{O_i}_{\tilde{O}_i} \neq 0$ and $m^{O_i}_{O_j} = 0$, we can derive $\mathrm{R}(O_j, O_i | \tilde{O}_i) \not\perp\!\!\!\perp \tilde{O}_i$ based on Lem. 3.

2. Suppose $\mathrm{An}_{\mathbf{O}}(O_i) \subseteq \mathbf{K}$, then $\mathrm{An}_{\mathbf{L}}(O_i) \nsubseteq \mathbf{J}$. Let $L_i \in \mathrm{An}_{\mathbf{L}}(O_i) \cap (\mathbf{L} \backslash \mathbf{J})$, there are two possible sub-cases as follows.

   (a) Suppose $\mathrm{De}_{\mathbf{O}}(O_i) = \emptyset$, let $O_j \in \mathrm{De}_{\mathbf{O}}(L_i) \backslash \{O_i\}$. An illustrative example is shown as Fig. 6(b). As $m^{L_i}_{O_i} m^{L_i}_{O_j} \neq 0$, $\mathrm{Cov}(O_j, \tilde{O}_i) \neq 0$ based on Lem. 4. Also, it is trivial that $\mathrm{Cov}(O_i, \tilde{O}_i) \neq 0$. Since $m^{O_i}_{O_i} m^{O_i}_{\tilde{O}_i} \neq 0$ and $m^{O_i}_{O_j} = 0$, we can derive $\mathrm{R}(O_j, O_i | \tilde{O}_i) \not\perp\!\!\!\perp \tilde{O}_i$ based on Lem. 3.

   (b) Suppose $\mathrm{De}_{\mathbf{O}}(O_i) \neq \emptyset$, let $O_j \in \mathrm{HSu}(L_i)$, it is trivial that $O_j \notin \mathrm{De}(O_i) \cup \{O_i\}$. An illustrative example is shown as Fig. 6(c). As $m^{L_i}_{O_i} m^{L_i}_{O_j} \neq 0$, $\mathrm{Cov}(O_j, \tilde{O}_i) \neq 0$ based on Lem. 4. Also, it is trivial that $\mathrm{Cov}(O_i, \tilde{O}_i) \neq 0$. Since $m^{O_i}_{O_i} m^{O_i}_{\tilde{O}_i} \neq 0$ and $m^{O_i}_{O_j} = 0$, we can derive $\mathrm{R}(O_j, O_i | \tilde{O}_i) \not\perp\!\!\!\perp \tilde{O}_i$ based on Lem. 3.

This finishes the proof. □

**Corollary 1.** Suppose $O_i$ satisfies Thm. 1, then $\forall O_j \in \mathbf{O} \backslash (\mathbf{K} \cup \{O_i\})$, $m^{O_i}_{O_j} = \frac{\mathrm{Cov}(\tilde{O}_i, O_j)}{\mathrm{Cov}(\tilde{O}_i, O_i)}$.

*Proof.* Based on Eqs. (15) and (16),

$$\tilde{O}_i = \sum_{O_{k_2} \in \mathbf{K}_2} \lambda_{ik_2} \epsilon_{O_{k_2}} + \epsilon_{O_i}. \tag{21}$$

$$O_j = \sum_{L_i \in \mathbf{J}} m_{O_j}^{L_i} \epsilon_{L_i} + \sum_{L_j \in \mathbf{L} \setminus \mathbf{J}} m_{O_j}^{L_j} \epsilon_{L_j} + \sum_{O_{k_1} \in \mathbf{K}_1} m_{O_j}^{O_{k_1}} \epsilon_{O_{k_1}} + \sum_{O_l \in \mathbf{O} \setminus \mathbf{K}} m_{O_j}^{O_l} \epsilon_{O_l}, \tag{22}$$

$$O_i = \sum_{L_i \in \mathbf{J}} m_{O_i}^{L_i} \epsilon_{L_i} + \sum_{O_{k_1} \in \mathbf{K}_1} m_{O_i}^{O_{k_1}} \epsilon_{O_{k_1}} + \epsilon_{O_i}, \tag{23}$$

Therefore,

$$\frac{\mathrm{Cov}(\tilde{O}_i, O_j)}{\mathrm{Cov}(\tilde{O}_i, O_i)} = \frac{m_{O_j}^{O_i} \mathrm{Var}(\epsilon_{O_i})}{\mathrm{Var}(\epsilon_{O_i})} = m_{O_j}^{O_i}. \tag{24}$$

$\square$

**Corollary 2.** Suppose $O_i$ satisfies Thm. 1, if we update $\mathbf{K}$ to $\mathbf{K} \cup \{O_i\}$ and $\tilde{O}_j$ to $\tilde{O}_j - m_{O_j}^{O_i} \tilde{O}_i$ for each $O_j \in \mathbf{O} \setminus \mathbf{K}$, Cond. 1 is still valid.

*Proof.* Based on Thm. 1, it is trivial that Cond. 1(1) is valid.

Based on Eq. (16), before update

$$\tilde{O}_i = \sum_{O_k \in \mathbf{K}_2} \lambda_{ik} \epsilon_{O_k} + \epsilon_{O_i}, \tag{25}$$

$$\tilde{O}_j = \sum_{L_i \in \mathbf{L} \setminus \mathbf{J}} m_{O_j}^{L_i} \epsilon_{L_i} + \sum_{O_k \in \mathbf{K}_2} \lambda_{jk} \epsilon_{O_k} + \sum_{O_l \in \mathbf{O} \setminus \mathbf{K}} m_{O_j}^{O_l} \epsilon_{O_l}, \tag{26}$$

then

$$\tilde{O}_j - m_{O_j}^{O_i} \tilde{O}_i = \sum_{L_i \in \mathbf{L} \setminus \mathbf{J}} m_{O_j}^{L_i} \epsilon_{L_i} + \sum_{O_k \in \mathbf{K}_2} \lambda'_{jk} \epsilon_{O_k} + \sum_{O_l \in \mathbf{O} \setminus (\mathbf{K} \cup \{O_i\})} m_{O_j}^{O_l} \epsilon_{O_l}, \tag{27}$$

where $\lambda'_{jk} = \lambda_{jk} - m_{O_j}^{O_i} \lambda_{ik}$, so Cond. 1(2) is also valid. $\square$

**Lemma 5.** *If* $\forall O \in \mathbf{O} \setminus \mathbf{K}, \mathrm{An}(O) \nsubseteq \mathbf{J} \cup \mathbf{K}$, *then* $\forall O \in \mathbf{O} \setminus \mathbf{K}, \mathrm{An}_{\mathbf{L}}(O) \nsubseteq \mathbf{J}$.

*Proof.* We prove it by contradiction. Suppose $\exists O_i \in \mathbf{O} \setminus \mathbf{K}$ s.t. $\mathrm{An}_{\mathbf{L}}(O_i) \subseteq \mathbf{J}$, then since $\mathrm{An}(O) \nsubseteq \mathbf{J} \cup \mathbf{K}, \mathrm{An}_{\mathbf{O}}(O_i) \nsubseteq \mathbf{K}$. Let $O_j \in \mathrm{An}_{\mathbf{O}}(O_i) \setminus \mathbf{K}$ s.t. $\mathrm{An}_{\mathbf{O}}(O_j) \cap (\mathrm{An}_{\mathbf{O}}(O_i) \setminus \mathbf{K}) = \emptyset$, so $\mathrm{An}_{\mathbf{O}}(O_j) \subseteq \mathbf{K}$. In addition, $\mathrm{An}_{\mathbf{L}}(O_j) \subseteq \mathrm{An}_{\mathbf{L}}(O_i) \subseteq \mathbf{J}$, so $\mathrm{An}(O_j) \subseteq \mathbf{J} \cup \mathbf{K}$, which leads to contradiction. $\square$

**Theorem 2.** Suppose $\forall O \in \mathbf{O} \setminus \mathbf{K}, \mathrm{An}(O) \nsubseteq \mathbf{J} \cup \mathbf{K}$. Given $O_i \in \mathbf{O} \setminus \mathbf{K}$, then $\mathrm{Ch}(O_i) = \emptyset$, $\mathrm{Pa}_{\mathbf{O}}(O_i) \setminus \mathbf{K} = \emptyset$, $|\mathrm{Pa}_{\mathbf{L}}(O_i) \setminus \mathbf{J}| = 1$, and $\mathrm{An}(\mathrm{Pa}_{\mathbf{L}}(O_i) \setminus \mathbf{J}) \subseteq \mathbf{J}$ if and only if $\forall \{O_j, O_k\} \subseteq \mathbf{O} \setminus (\mathbf{K} \cup \{O_i\})$ where $\mathrm{Cov}(\tilde{O}_i, O_j) \mathrm{Cov}(\tilde{O}_i, O_k) \neq 0$, $\mathrm{R}(O_j, O_k | \tilde{O}_i) \perp\!\!\!\perp \tilde{O}_i$.

**Proof sketch.** If $O_i$ satisfies the graphical condition, we can prove independence based on Lem. 2 (let $\mathrm{Pa}_{\mathbf{L}}(O_i) \setminus \mathbf{J} = \{L\}$, $\epsilon_L$ serves as $e$ in Eq. (13)). Otherwise, there exists $V \in \mathbf{V} \setminus (\mathbf{J} \cup \mathbf{K})$ and $\{O_j, O_k\} \subseteq \mathbf{O} \setminus (\mathbf{K} \cup \{O_i\})$ s.t. $\mathrm{Cov}(\tilde{O}_i, O_j) \mathrm{Cov}(\tilde{O}_i, O_k) \neq 0$, $m_{\tilde{O}_i}^V m_{O_j}^V \neq 0$, and $m_{O_k}^V = 0$, so dependence can be proven by Lem. 3.

*Proof.* "Only if": Let $\mathrm{Pa}_{\mathbf{L}}(O_i) \setminus \mathbf{J} = \{L_i\}$, note that $\{O_j, O_k\} \cap \mathrm{De}(O_i) = \emptyset$, based on Eqs. (15) and (16),

$$\tilde{O}_i = m_{O_i}^{L_i} \epsilon_{L_i} + \sum_{O_{l_2} \in \mathbf{K}_2} \lambda_{il_2} \epsilon_{O_{l_2}} + \epsilon_{O_i}. \tag{28}$$

$$O_j = \sum_{L_j \in \mathbf{J}} m_{O_j}^{L_j} \epsilon_{L_j} + \sum_{L_k \in \mathbf{L} \setminus \mathbf{J}} m_{O_j}^{L_k} \epsilon_{L_k} + \sum_{O_{l_1} \in \mathbf{K}_1} m_{O_j}^{O_{l_1}} \epsilon_{O_{l_1}} + \sum_{O_m \in \mathbf{O} \setminus (\mathbf{K} \cup \{O_i\})} m_{O_j}^{O_m} \epsilon_{O_m}, \tag{29}$$

$$O_k = \sum_{L_j \in \mathbf{J}} m_{O_k}^{L_j} \epsilon_{L_j} + \sum_{L_k \in \mathbf{L} \setminus \mathbf{J}} m_{O_k}^{L_k} \epsilon_{L_k} + \sum_{O_{l_1} \in \mathbf{K}_1} m_{O_k}^{O_{l_1}} \epsilon_{O_{l_1}} + \sum_{O_m \in \mathbf{O} \setminus (\mathbf{K} \cup \{O_i\})} m_{O_k}^{O_m} \epsilon_{O_m}. \tag{30}$$

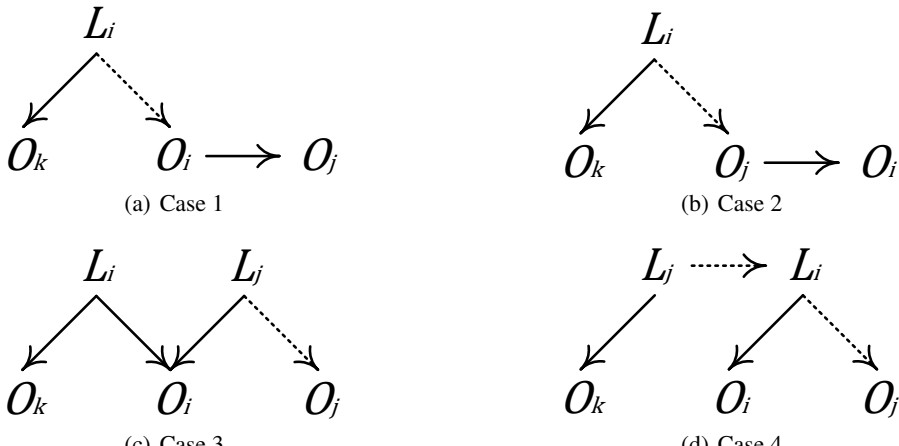

(a) Case 1      (b) Case 2

(c) Case 3      (d) Case 4

Figure 7: Illustration of "If" part in proof of Thm. 2. A dotted arrow from $V_1$ to $V_2$ means that $V_2 \in \mathrm{De}(V_1)$.

Since $\mathrm{Cov}(\tilde{O}_i, O_j)\mathrm{Cov}(\tilde{O}_i, O_k) = (m_{O_i}^{L_i})^2 m_{O_j}^{L_i} m_{O_k}^{L_i} (\mathrm{Var}(\epsilon_{L_i}))^2 \neq 0$, there is $m_{O_j}^{L_i} m_{O_k}^{L_i} \neq 0$. Based on Lem. 2, $\mathrm{R}(O_j, O_k | \tilde{O}_i) \perp\!\!\!\perp \tilde{O}_i$, where $\epsilon_{L_i}$ serves as $e$ in Eq. (13).

"If": We prove this part by contradiction.

1. Suppose $\mathrm{Ch}(O_i) \neq \emptyset$, let $O_j \in \mathrm{Ch}(O_i)$. Based on Lem. 5, $\mathrm{An}_{\mathbf{L}}(O_i) \not\subseteq \mathbf{J}$ and let $L_i \in \mathrm{An}_{\mathbf{L}}(O_i)\backslash\mathbf{J}$. Besides, let $O_k \in \mathrm{HSu}(L_i)$, it is trivial that $O_k \notin \{O_i, O_j\}$ and $O_i \notin \mathrm{An}(O_k)$. An illustrative example is shown as Fig. 7(a). As $m_{O_i}^{L_i} m_{O_j}^{L_i} m_{O_k}^{L_i} \neq 0$, $\mathrm{Cov}(O_j, \tilde{O}_i)\mathrm{Cov}(O_k, \tilde{O}_i) \neq 0$ based on Lem. 4. Since $m_{\tilde{O}_i}^{O_i} m_{O_j}^{O_i} \neq 0$ and $m_{O_k}^{O_i} = 0$, we can derive $\mathrm{R}(O_j, O_k | \tilde{O}_i) \not\perp\!\!\!\perp \tilde{O}_i$ based on Lem. 3.

2. Suppose $\mathrm{Ch}(O_i) = \emptyset$ and $\mathrm{Pa}_{\mathbf{O}}(O_i)\backslash\mathbf{K} \neq \emptyset$, let $O_j \in \mathrm{Pa}_{\mathbf{O}}(O_i)\backslash\mathbf{K}$. Based on Lem. 5, $\mathrm{An}_{\mathbf{L}}(O_j) \not\subseteq \mathbf{J}$ and let $L_i \in \mathrm{An}_{\mathbf{L}}(O_j)\backslash\mathbf{J}$. Besides, let $O_k \in \mathrm{HSu}(L_i)$, it is trivial that $O_k \notin \{O_i, O_j\}$ and $O_j \notin \mathrm{An}(O_k)$. An illustrative example is shown as Fig. 7(b). As $m_{O_i}^{L_i} m_{O_j}^{L_i} m_{O_k}^{L_i} \neq 0$, $\mathrm{Cov}(O_j, \tilde{O}_i)\mathrm{Cov}(O_k, \tilde{O}_i) \neq 0$ based on Lem. 4. Since $m_{\tilde{O}_i}^{O_j} m_{O_j}^{O_j} \neq 0$ and $m_{O_k}^{O_j} = 0$, we can derive $\mathrm{R}(O_j, O_k | \tilde{O}_i) \not\perp\!\!\!\perp \tilde{O}_i$ based on Lem. 3.

3. Suppose $\mathrm{Ch}(O_i) = \mathrm{Pa}_{\mathbf{O}}(O_i)\backslash\mathbf{K} = \emptyset$ and $|\mathrm{Pa}_{\mathbf{L}}(O_i)\backslash\mathbf{J}| \geq 2$, let $\{L_i, L_j\} \subseteq \mathrm{Pa}_{\mathbf{L}}(O_i)\backslash\mathbf{J}$. Without loss of generality, let $L_j \notin \mathrm{An}(L_i)$. Let $O_j \in \mathrm{De}(L_j)\backslash\{O_i\}$ and $O_k \in \mathrm{HSu}(L_i)$. It is trivial that $O_k \notin \{O_i, O_j\}$ and $L_j \notin \mathrm{An}(O_k)$. An illustrative example is shown as Fig. 7(c). As $m_{O_j}^{L_j} m_{O_i}^{L_j} \neq 0$ and $m_{O_k}^{L_i} m_{O_i}^{L_i} \neq 0$, $\mathrm{Cov}(O_j, \tilde{O}_i)\mathrm{Cov}(O_k, \tilde{O}_i) \neq 0$ based on Lem. 4. Since $m_{\tilde{O}_i}^{L_j} m_{O_j}^{L_j} \neq 0$ and $m_{O_k}^{L_j} = 0$, we can derive $\mathrm{R}(O_j, O_k | \tilde{O}_i) \not\perp\!\!\!\perp \tilde{O}_i$ based on Lem. 3.

4. Suppose $\mathrm{Ch}(O_i) = \mathrm{Pa}_{\mathbf{O}}(O_i)\backslash\mathbf{K} = \emptyset$, $|\mathrm{Pa}_{\mathbf{L}}(O_i)\backslash\mathbf{J}| = 1$, and $\mathrm{An}(\mathrm{Pa}_{\mathbf{L}}(O_i)\backslash\mathbf{J}) \not\subseteq \mathbf{J}$. Let $\mathrm{Pa}_{\mathbf{L}}(O_i)\backslash\mathbf{J} = \{L_i\}$, $L_j \in \mathrm{An}(L_i)\backslash\mathbf{J}$, $O_j \in \mathrm{De}(L_i)\backslash\{O_i\}$, and $O_k \in \mathrm{HSu}(L_j)$. It is trivial that $O_k \notin \{O_i, O_j\}$ and $L_i \notin \mathrm{An}(O_k)$. An illustrative example is shown as Fig. 7(d). As $m_{O_i}^{L_j} m_{O_j}^{L_j} m_{O_k}^{L_j} \neq 0$, $\mathrm{Cov}(O_j, \tilde{O}_i)\mathrm{Cov}(O_k, \tilde{O}_i) \neq 0$ based on Lem. 4. Since $m_{\tilde{O}_i}^{L_i} m_{O_j}^{L_i} \neq 0$ and $m_{O_k}^{L_i} = 0$, we can derive $\mathrm{R}(O_j, O_k | \tilde{O}_i) \not\perp\!\!\!\perp \tilde{O}_i$ based on Lem. 3.

This finishes the proof. $\qquad\qquad\square$

**Proposition 1.** Suppose $O_i$ and $O_j$ satisfy Thm. 2, then $\mathrm{Pa}_{\mathbf{L}}(O_i)\backslash\mathbf{J} = \mathrm{Pa}_{\mathbf{L}}(O_j)\backslash\mathbf{J}$ if and only if $\mathrm{Cov}(\tilde{O}_i, O_j) \neq 0$.

*Proof.* Let $\mathrm{Pa}_{\mathbf{L}}(O_i)\backslash\mathbf{J} = \{L_i\}$ and $\mathrm{Pa}_{\mathbf{L}}(O_j)\backslash\mathbf{J} = \{L_j\}$. Based on Eqs. (15) and (16),

$$\tilde{O}_i = m_{O_i}^{L_i}\epsilon_{L_i} + \sum_{O_{l_2}\in\mathbf{K}_2}\lambda_{il_2}\epsilon_{O_{l_2}} + \epsilon_{O_i}. \tag{31}$$

$$O_j = \sum_{L_k\in\mathbf{J}}m_{O_j}^{L_k}\epsilon_{L_k} + m_{O_j}^{L_j}\epsilon_{L_j} + \sum_{O_{l_1}\in\mathbf{K}_1}m_{O_j}^{O_{l_1}}\epsilon_{O_{l_1}} + \epsilon_{O_j}. \tag{32}$$

Obviously, $L_i = L_j$ if and only if $\mathrm{Cov}(\tilde{O}_i, O_j) = m_{O_i}^{L_i}m_{O_j}^{L_j}\mathrm{Cov}(\epsilon_{L_i}, \epsilon_{L_j}) \neq 0$, that is, $L_i = L_j$. □

**Proposition 2.** Suppose $O_i$ satisfies Thm. 2, then $O_i \in \mathrm{HSu}(\mathrm{Pa}_{\mathbf{L}}(O_i)\backslash\mathbf{J})$ if and only if $\forall O_j$ satisfying Thm. 2 and $\mathrm{Pa}_{\mathbf{L}}(O_j)\backslash\mathbf{J} = \mathrm{Pa}_{\mathbf{L}}(O_i)\backslash\mathbf{J}$, $\|\mathbf{M}_{\{O_i\}}^{\mathbf{J}}\|_0 \leq \|\mathbf{M}_{\{O_j\}}^{\mathbf{J}}\|_0$.

*Proof.* Let $\mathrm{Pa}_{\mathbf{L}}(O_i)\backslash\mathbf{J} = \{L_i\}$.

"Only if". For each $O_j$ satisfying $O_j \in \mathrm{De}(L_i)$, there is $\mathrm{An}(O_j) \cap \mathbf{J} \supseteq \mathrm{An}(L_i) \cap \mathbf{J}$. Since $O_i \in \mathrm{HSu}(L_i)$, $\mathrm{An}(L_i) \cap \mathbf{J} = \mathrm{An}(O_i) \cap \mathbf{J}$. Therefore, $\|\mathbf{M}_{\{O_i\}}^{\mathbf{J}}\|_0 \leq \|\mathbf{M}_{\{O_j\}}^{\mathbf{J}}\|_0$.

"If". We prove this part by contradiction. Suppose $O_i \notin \mathrm{HSu}(L_i)$. Let $O_j \in \mathrm{HSu}(L_i)$, then we have $\|\mathbf{M}_{\{O_i\}}^{\mathbf{J}}\|_0 \leq \|\mathbf{M}_{\{O_j\}}^{\mathbf{J}}\|_0 = |\mathrm{An}(L_i) \cap \mathbf{J}|$. Since $O_i \in \mathrm{De}(L_i)$, $\|\mathbf{M}_{\{O_i\}}^{\mathbf{J}}\|_0 \geq |\mathrm{An}(L_i) \cap \mathbf{J}|$. Therefore, $\|\mathbf{M}_{\{O_i\}}^{\mathbf{J}}\|_0 = \|\mathbf{M}_{\{O_j\}}^{\mathbf{J}}\|_0$, that is, $\mathrm{An}(O_i) \cap \mathbf{J} = \mathrm{An}(O_j) \cap \mathbf{J} = \mathrm{An}(L_i)$, note that $O_i$ satisfies Thm. 2, so $O_i \in \mathrm{HSu}(L_i)$, this leads to contradiction. □

**Definition 3.** (Cumulant) Given $n$ random variables $V_1, ..., V_n$, the $k$-th order cumulant is defined as a tensor of size $n \times ... \times n$ ($k$ times), whole element at position $(i_1, ..., i_k)$ is

$$\mathrm{Cum}(V_{i_1}, ..., V_{i_k}) = \sum_{\pi}(-1)^{|\pi|-1}(|\pi| - 1)!\prod_{B\in\pi}\mathbb{E}\left[\prod_{j\in B}V_j\right], \tag{33}$$

where $\pi$ is enumerated over all partitions of $\{i_1, ..., i_k\}$.

**Corollary 3.** Suppose $O_i$ satisfies Thm. 2 and $O_i \in \mathrm{HSu}(L_i)$, then $\forall O_j \in \mathbf{O}\backslash(\mathbf{K} \cup \{O_i\})$,

$$m_{O_i}^{L_i}m_{O_j}^{L_i} = \mathrm{Cov}(\tilde{O}_i, O_j), \quad \left(\frac{m_{O_i}^{L_i}}{m_{O_j}^{L_i}}\right)^2 = \frac{\mathrm{Cum}(\tilde{O}_i, \tilde{O}_i, \tilde{O}_i, O_j)}{\mathrm{Cum}(\tilde{O}_i, O_j, O_j, O_j)}. \tag{34}$$

*Proof.* Note that $O_j \notin \mathrm{De}(O_i)$, based on Eqs. (15) and (16),

$$\tilde{O}_i = m_{O_i}^{L_i}\epsilon_{L_i} + \sum_{O_{l_2}\in\mathbf{K}_2}\lambda_{il_2}\epsilon_{O_{l_2}} + \epsilon_{O_i}. \tag{35}$$

$$O_j = \sum_{L_j\in\mathbf{J}}m_{O_j}^{L_j}\epsilon_{L_j} + \sum_{L_k\in\mathbf{L}\backslash\mathbf{J}}m_{O_j}^{L_k}\epsilon_{L_k} + \sum_{O_{l_1}\in\mathbf{K}_1}m_{O_j}^{O_{l_1}}\epsilon_{O_{l_1}} + \sum_{O_m\in\mathbf{O}\backslash(\mathbf{K}\cup\{O_i\})}m_{O_j}^{O_m}\epsilon_{O_m}. \tag{36}$$

Clearly,

$$\mathrm{Cov}(\tilde{O}_i, O_j) = m_{O_i}^{L_i}m_{O_j}^{L_i}\mathrm{Var}(\epsilon_{L_i}) = m_{O_i}^{L_i}m_{O_j}^{L_i}. \tag{37}$$

The second equation holds because we assume the exogenous noise of each latent variable has unit variance.

Because we assume each exogenous noise has zero mean, $\mathbb{E}[\tilde{O}_i] = \mathbb{E}[O_j] = 0$. There is

$$\mathrm{Cum}(\tilde{O}_i, \tilde{O}_i, \tilde{O}_i, O_j) = \mathbb{E}[\tilde{O}_i^3 O_j] - 3\mathbb{E}[\tilde{O}_i O_j]\mathbb{E}[\tilde{O}_i^2]. \tag{38}$$

Let $\tilde{e}_i = \tilde{O}_i - m^{L_i}_{O_i}\epsilon_{L_i}$ and $e_j = O_j - m^{L_i}_{O_j}\epsilon_{L_i}$. It follows that $\epsilon_{L_i}, \tilde{e}_i, e_j$ are mutually independent. Therefore,

$$
\begin{aligned}
&\mathrm{Cum}(\tilde{O}_i, \tilde{O}_i, \tilde{O}_i, O_j) \\
&= \underbrace{\mathbb{E}[(m^{L_i}_{O_i}\epsilon_{L_i} + \tilde{e}_i)^3(m^{L_i}_{O_j}\epsilon_{L_i} + e_j)]}_{\mathbb{E}[\tilde{O}_i^3 O_j]} - 3\underbrace{\mathbb{E}[(m^{L_i}_{O_i}\epsilon_{L_i} + \tilde{e}_i)(m^{L_i}_{O_j}\epsilon_{L_i} + e_j)]}_{\mathbb{E}[\tilde{O}_i O_j]}\underbrace{\mathbb{E}[(m^{L_i}_{O_i}\epsilon_{L_i} + \tilde{e}_i)^2]}_{\mathbb{E}[\tilde{O}_i^2]} \\
&= \underbrace{(m^{L_i}_{O_i})^3 m^{L_i}_{O_j}\mathbb{E}[\epsilon_{L_i}^4] + 3m^{L_i}_{O_i}m^{L_i}_{O_j}\mathbb{E}[\epsilon_{L_i}^2\tilde{e}_i^2]}_{\mathbb{E}[(m^{L_i}_{O_i}\epsilon_{L_i}+\tilde{e}_i)^3(m^{L_i}_{O_j}\epsilon_{L_i}+e_j)]} - 3\underbrace{\left((m^{L_i}_{O_i})^3 m^{L_i}_{O_j}(\mathbb{E}[\epsilon_{L_i}^2])^2 + m^{L_i}_{O_i}m^{L_i}_{O_j}\mathbb{E}[\epsilon_{L_i}^2]\mathbb{E}[\tilde{e}_i^2]\right)}_{\mathbb{E}[(m^{L_i}_{O_i}\epsilon_{L_i}+\tilde{e}_i)(m^{L_i}_{O_j}\epsilon_{L_i}+e_j)]\mathbb{E}[(m^{L_i}_{O_i}\epsilon_{L_i}+\tilde{e}_i)^2]} \\
&= (m^{L_i}_{O_i})^3 m^{L_i}_{O_j}(\mathbb{E}[\epsilon_{L_i}^4] - 3(\mathbb{E}[\epsilon_{L_i}^2])^2). 
\end{aligned}
\tag{39}
$$

The second equation holds because for any three independent random variables $V_1, V_2, V_3$ with $\mathbb{E}[V_3] = 0$

$$\mathbb{E}[V_1^2 V_2 V_3] = \mathbb{E}[V_1^2]\mathbb{E}[V_2]\mathbb{E}[V_3] = 0, \tag{40}$$

$$\mathbb{E}[V_1^3 V_3] = \mathbb{E}[V_1^3]\mathbb{E}[V_3] = 0, \tag{41}$$

$$\mathbb{E}[V_1 V_3] = \mathbb{E}[V_1]\mathbb{E}[V_3] = 0. \tag{42}$$

The third equation holds because

$$\mathbb{E}[\epsilon_{L_i}^2 \tilde{e}_i^2] = \mathbb{E}[\epsilon_{L_i}^2]\mathbb{E}[\tilde{e}_i^2]. \tag{43}$$

Similarly,

$$\mathrm{Cum}(\tilde{O}_i, O_j, O_j, O_j) = m^{L_i}_{O_i}(m^{L_i}_{O_j})^3(\mathbb{E}[\epsilon_{L_i}^4] - 3(\mathbb{E}[\epsilon_{L_i}^2])^2). \tag{44}$$

Therefore,

$$\left(\frac{m^{L_i}_{O_i}}{m^{L_i}_{O_j}}\right)^2 = \frac{\mathrm{Cum}(\tilde{O}_i, \tilde{O}_i, \tilde{O}_i, O_j)}{\mathrm{Cum}(\tilde{O}_i, O_j, O_j, O_j)}. \tag{45}$$

$\square$

**Remark.** In the above proof, we implicitly assume that $\mathbb{E}[\epsilon_{L_i}^4] \neq 3(\mathbb{E}[\epsilon_{L_i}^2])^2$, that is, the excess kurtosis of $\epsilon_{L_i}$ is not zero. An excess kurtosis of zero indicates that a distribution has identical tail behavior and peak characteristics as a Gaussian distribution, which means the probability of extreme values occurring is precisely equivalent to that of a Gaussian distribution. Considering that $\epsilon_{L_i}$ is a non-Gaussian variable, this is a mild technical assumption.

**Corollary 4.** Suppose $O_i$ satisfies Thm. 2 and $O_i \in \mathrm{HSu}(L_i)$, if we update $\mathbf{J}$ to $\mathbf{J} \cup \{L_i\}$, $\mathbf{K}$ to $\mathbf{K} \cup \{O_i\}$, and $\tilde{O}_j$ to $\tilde{O}_j - (m^{L_i}_{O_j}/m^{L_i}_{O_i})\tilde{O}_i$ for each $O_j \in \mathbf{O}\backslash\mathbf{K}$, Cond. 1 is still valid.

*Proof.* Based on Thm. 2, it is trivial that Cond. 1(1) is valid.

Note that $O_j \notin \mathrm{De}(O_i)$, based on Eq. (16), before removal

$$\tilde{O}_i = m^{L_i}_{O_i}\epsilon_{L_i} + \sum_{O_k \in \mathbf{K}_2} \lambda_{ik}\epsilon_{O_k} + \epsilon_{O_i}. \tag{46}$$

$$\tilde{O}_j = \sum_{L_j \in \mathbf{L}\backslash\mathbf{J}} m^{L_j}_{O_j}\epsilon_{L_j} + \sum_{O_k \in \mathbf{K}_2} \lambda_{jk}\epsilon_{O_k} + \sum_{O_l \in \mathbf{O}\backslash(\mathbf{K}\cup\{O_i\})} m^{O_l}_{O_j}\epsilon_{O_l}, \tag{47}$$

then

$$\tilde{O}_j - \frac{m^{L_i}_{O_j}}{m^{L_i}_{O_i}}\tilde{O}_i = \sum_{L_j \in \mathbf{L}\backslash(\mathbf{J}\cup\{L_i\})} m^{L_j}_{O_j}\epsilon_{L_j} + \sum_{O_k \in \mathbf{K}_2\cup\{O_i\}} \lambda'_{jk}\epsilon_{O_k} + \sum_{O_l \in \mathbf{O}\backslash(\mathbf{K}\cup\{O_i\})} m^{O_l}_{O_j}\epsilon_{O_l}, \tag{48}$$

where $\lambda'_{jk} = \lambda_{jk} - \frac{m^{L_i}_{O_j}}{m^{L_i}_{O_i}}\lambda_{ik}$ if $O_k \in \mathbf{K}_2$ and $\lambda'_{jk} = -\frac{m^{L_i}_{O_j}}{m^{L_i}_{O_i}}$ if $O_k = O_i$, so Cond. 1(2) is also valid. $\square$

**Theorem 3.** Suppose the observed variables are generated by a LiNGAM with latent variables satisfying Asmp. 1, in the limit of infinite data, Alg. 1 identifies latent variables and ancestral relationships correctly.

*Proof.* Based on Thm. 1, Cors. 1 and 2, Alg. 1 correctly estimates $\mathbf{M_O^O}$. Based on Thm. 2, Props. 1 and 2, Cors. 3 and 4, Alg. 1 correctly identifies latent variables by locating their respective homologous surrogates and estimates $\mathbf{M_O^L}$. Furthermore, given any two latent variable $L_i, L_j$, Alg. 1 correctly determines whether $L_i$ is an ancestor of $L_j$ by checking whether $L_i$ is an ancestor of $L_j$'s homologous surrogates. $\square$

### C.3 PROOF OF THEORETICAL RESULTS IN SEC. 4.

**Definition 4.** (Generalized homologous surrogate) $O \in \mathbf{O}$ is called a generalized homologous surrogate of $L \in \mathbf{L}$, denoted by $O \in \text{GHSu}(L)$, if $O \in \text{Ch}(L)$ and $\text{Pa}_{\mathbf{L}}(O) \subseteq \text{An}(L) \cup \{L\}$.

**Assumption 2.** Asmp. 1 holds and $\forall\{L_i, L_j\} \subseteq \mathbf{L}$ where $L_i \in \text{An}(L_j)$, $\exists\{O_{j_1}, O_{j_2}\} \subseteq \text{GHSu}(L_j)$ s.t $O_{j_1} \notin \text{Ch}(L_i)$ and $O_{j_2} \notin \text{Ch}(L_i)$.

As mentioned in the main text, we can remove causal relations between observed variables given $\mathbf{A_O^O}$. Specifically, for each $O_i \in \mathbf{O}$, we let

$$O_i^* = O_i - \sum_{O_j \in \text{Pa}(O_i)} a_{O_i}^{O_j} O_j. \tag{49}$$

We denote by $\mathcal{G}^*$ the causal graph among $\mathbf{L} \cup \mathbf{O}^*$ where $\mathbf{O}^* = \{O_i^*\}_i$. Also, we use $\text{Pa}^*(V)$ to denote $V$'s parents in $\mathcal{G}^*$.

**Lemma 6.** *Given $L \in \mathbf{L}$ and $O \in \mathbf{O}$, $O \in \text{GHSu}(L)$ if and only if $\text{An}^*(O^*) = \text{An}^*(L) \cup \{L\}$.*

*Proof.* This can be readily derived from the definition of generalized homologous surrogates. $\square$

**Lemma 1.** *$\forall L_i \in \mathbf{L}$ and $O_i \in \mathbf{O}$, $O_i \in \text{GHSu}(L_i)$ if and only if $m_{O_i^*}^{L_i} \neq 0$ and $\forall O_j \in \mathbf{O}$ where $m_{O_j^*}^{L_i} \neq 0$, $\|\mathbf{M}_{\{O_i^*\}}^{\mathbf{L}}\|_0 \leq \|\mathbf{M}_{\{O_j^*\}}^{\mathbf{L}}\|_0$. Besides, there is $a_{O_i}^{L_i} = m_{O_i^*}^{L_i}$.*

*Proof.* "Only if". For each $O_j^*$ where $m_{O_j^*}^{L_i} \neq 0$, there is $\text{An}^*(O_j^*) \supseteq \text{An}^*(L_i) \cup \{L_i\}$. Based on Lem. 6, $\text{An}^*(O_i^*) = \text{An}^*(L_i) \cup \{L_i\}$, so $\|\mathbf{M}_{\{O_i^*\}}^{\mathbf{L}}\|_0 \leq \|\mathbf{M}_{\{O_j^*\}}^{\mathbf{L}}\|_0$.

"If". We prove this part by contradiction. Suppose $O_i \notin \text{GHSu}(L_i)$. Let $O_j \in \text{GHSu}(L_i)$, then $\|\mathbf{M}_{\{O_i^*\}}^{\mathbf{L}}\|_0 \leq \|\mathbf{M}_{\{O_j^*\}}^{\mathbf{L}}\|_0 = |\text{An}^*(L_i) \cup \{L_i\}|$ based on Lem. 6. Since $m_{O_i^*}^{L_i} \neq 0$, $O_i^* \in \text{De}^*(L_i)$, that is, $\|\mathbf{M}_{\{O_i^*\}}^{\mathbf{L}}\|_0 \geq |\text{An}^*(L_i) \cup \{L_i\}|$. Therefore, $\|\mathbf{M}_{\{O_i^*\}}^{\mathbf{L}}\|_0 = \|\mathbf{M}_{\{O_j^*\}}^{\mathbf{L}}\|_0$, that is, $\text{An}^*(O_i^*) = \text{An}^*(L_i) \cup \{L_i\}$, so $O_i \in \text{GHSu}(L_i)$ based on Lem. 6, this leads to contradiction.

Finally, based on Lem. 6, it is trivial that if $O_i \in \text{GHSu}(L_i)$, $a_{O_i}^{L_i} = m_{O_i^*}^{L_i}$ because there is only one directed path from $L_i$ to $O_i^*$ in $\mathcal{G}^*$, which is exactly $L_i \to O_i^*$. $\square$

**Theorem 4.** Suppose $\{L_i, L_j\} \subseteq \mathbf{L}$, $L_j \in \text{De}^n(L_i)$. $\forall O_j \in \text{GHSu}(L_j)$, let

$$\mu_{O_j^*}^{L_i} = m_{O_j^*}^{L_i} - \sum_{L_k \in \text{De}(L_i) \cap \text{An}(L_j)} m_{L_k}^{L_i} a_{O_j}^{L_k}. \tag{50}$$

(a) There exists $\{O_{j_1}, O_{j_2}\} \subseteq \text{GHSu}(L_j)$ s.t. $\mu_{O_{j_1}^*}^{L_i} / a_{O_{j_1}}^{L_j} = \mu_{O_{j_2}^*}^{L_i} / a_{O_{j_2}}^{L_j}$ and $m_{L_j}^{L_i} = \mu_{O_{j_1}^*}^{L_i} / a_{O_{j_1}}^{L_j}$.

(b) $a_{O_j}^{L_i} = \mu_{O_j^*}^{L_i} - m_{L_j}^{L_i} a_{O_j}^{L_j}$.

**Proof sketch.** We can derive that $\mu_{O_j^*}^{L_i} = a_{O_j}^{L_i} + m_{L_j}^{L_i} a_{O_j}^{L_j}$, so (b) holds naturally. For (a), based on the faithfulness assumption, $\mu_{O_{j_1}^*}^{L_i}/a_{O_{j_1}}^{L_j} = \mu_{O_{j_2}^*}^{L_i}/a_{O_{j_2}}^{L_j}$ if and only if $a_{O_{j_1}}^{L_i} = a_{O_{j_2}}^{L_i} = 0$, that is, $O_{j_1} \notin \mathrm{Ch}(L_j)$ and $O_{j_2} \notin \mathrm{Ch}(L_j)$. In this case, $\mu_{O_{j_1}^*}^{L_i}/a_{O_{j_1}}^{L_j} = m_{L_j}^{L_i}$ trivially.

*Proof.* For any $L_i \in \mathbf{L}$ and $O_j^* \in \mathbf{O}^*$, there is

$$m_{O_j^*}^{L_i} = a_{O_j}^{L_i} + \sum_{L_k \in \mathrm{De}^*(L_i) \cap \mathrm{Pa}^*(O_j^*)} m_{L_k}^{L_i} a_{O_j}^{L_k}. \tag{51}$$

Given $O_j \in \mathrm{GHSu}(L_j)$, based on Lem. 6, $\mathrm{Pa}^*(O_j^*) \subseteq \mathrm{An}^*(O_j^*) = \mathrm{An}^*(L_j) \cup \{L_j\}$, so $\big(\mathrm{De}^*(L_i) \cap \mathrm{Pa}^*(O_j^*)\big)\backslash\{L_j\} \subseteq \mathrm{De}^*(L_i) \cap \mathrm{An}^*(L_j) = \mathrm{De}(L_i) \cap \mathrm{An}(L_j)$. Because $m_L^{L_i} a_{O_j}^L = 0$ if $L \notin \mathrm{De}^*(L_i) \cap \mathrm{Pa}^*(O_j^*)$, we can rewrite Eq. (51) as

$$m_{O_j^*}^{L_i} = a_{O_j}^{L_i} + \sum_{L_k \in \mathrm{De}(L_i) \cap \mathrm{An}(L_j)} m_{L_k}^{L_i} a_{O_j}^{L_k} + m_{L_j}^{L_i} a_{O_j}^{L_j}. \tag{52}$$

Therefore,

$$\mu_{O_j^*}^{L_i} = m_{O_j^*}^{L_i} - \sum_{L_k \in \mathrm{De}(L_i) \cap \mathrm{An}(L_j)} m_{L_k}^{L_i} a_{O_j}^{L_k} = a_{O_j}^{L_i} + m_{L_j}^{L_i} a_{O_j}^{L_j}. \tag{53}$$

Let $\{O_{j_1}, O_{j_2}\} \subseteq \mathrm{GHSu}(L_j)$. On the one hand, if $O_{j_1} \notin \mathrm{Ch}(L_i)$ and $O_{j_2} \notin \mathrm{Ch}(L_i)$, then $a_{O_{j_1}}^{L_i} = a_{O_{j_2}}^{L_i} = 0$, so $\mu_{O_{j_1}^*}^{L_i}/a_{O_{j_1}}^{L_j} = \mu_{O_{j_2}^*}^{L_i}/a_{O_{j_2}}^{L_j} = m_{L_j}^{L_i}$. On the other hand, if $O_{j_1} \in \mathrm{Ch}(L_i)$ or $O_{j_2} \in \mathrm{Ch}(L_i)$, based on the faithfulness assumption, $\mu_{O_{j_1}^*}^{L_i}/a_{O_{j_1}}^{L_j} \neq \mu_{O_{j_2}^*}^{L_i}/a_{O_{j_2}}^{L_j}$. Therefore, (a) holds.

Besides, based on Eq. (53), it is trivial that (b) holds. □

**Theorem 5.** *Suppose the observed variables are generated by a LiNGAM with latent variables satisfying Asmp. 2, in the limit of infinite data, Algs. 1 and 2 together identifies latent variables and parental relationships correctly.*

*Proof.* Based on Thm. 3, Alg. 1 correctly identifies latent variables and estimates $\mathbf{M_L^O}$ and $\mathbf{M_O^O}$. Based on Thm. 4, Alg. 2 correctly estimates $\mathbf{M_L^L}$. Therefore, $\mathbf{M}$ is estimated correctly, from which $\mathbf{A}$ can be derived based on Eq. (5). □

## D  EXPERIMENT ON REAL-WORLD DATA

The Holzinger and Swineford 1939 dataset consists of mental ability test scores of seventh- and eighth-grade children from two different schools (Pasteur and Grant-White). There are 9 variables, which can be categorized into three dimensions: Visual $(O_1, O_2, O_3)$, Textual $(O_4, O_5, O_6)$, and Speeded $(O_6, O_7, O_8)$. The result returned by our algorithm is shown as Fig. 8. Our algorithm correctly identifies the textual factor while merges the visual factor and the speed factor into a single factor. This can be attributed to the fact that both the visual factor and speed factor depends on innate abilities, while the textual factor highly depends on learning experience.

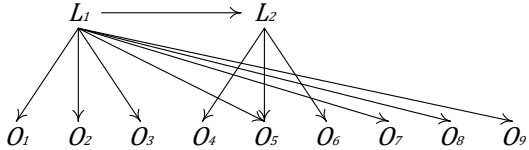

Figure 8: The output of our algorithm on the Holzinger and Swineford dataset.

