# OpenReview forum: "Recovery of Causal Graph Involving Latent Variables via Homologous Surrogates"
_ICLR.cc/2025/Conference — ICLR 2025 Poster_

### Official Review · Reviewer_B7bR · 2024-10-21

**Soundness:** 3
**Presentation:** 2
**Contribution:** 2
**Rating:** 6
**Confidence:** 3

**Summary:**

The paper proposes the use of homologous surrogates to identify latent variables and recover the causal graph. The main contribution of this work is that, unlike pure child variables, homologous surrogates allow for the presence of other parents. Based on this concept, the paper develops corresponding theoretical support and applies it to an algorithm.

**Strengths:**

The innovation of this paper is quite novel, as it uses homologous surrogates to identify latent variables and then recover the causal graph. The supporting theories are also rich, and the corresponding proofs are complete. The theories and propositions applied in the algorithm section are clearly explained, along with examples. These theories are crucial for the implementation of the algorithm. The method proposed in this paper offers a new perspective for discovering latent variables.

**Weaknesses:**

In the third chapter, PARTIAL RECOVERY, the explanation of the algorithm and theoretical part is not very clear in terms of formatting. The formulas and theoretical support needed for the preceding theories and hypotheses are instead explained afterward, which increases the difficulty of reading and affects the logical flow. For example, in "Identifying Observed Root Variables.", in the "Intuition" section, Theorems 1 and 2 are mentioned, so they should be placed before the "Intuition" section. The theories used in the proof should appear before the proof itself, rather than afterwards.

**Questions:**

1. In Algorithm 1, the font of "Thm.1" in line 3 is different from the "Thm.1" in other parts of the pseudocode, which needs attention.
2. For Algorithm 1 and Algorithm 2, the authors use Figure 2 and Figure 3 for explanation, but the tracking example is placed in the figure captions, and the corresponding steps are not explained in detail. It is recommended to add more detailed explanations in the main text to guide readers in understanding the example presented in the figures.
3. Please add your source code link to the paper.

---

> ### Author Response · Authors · 2024-11-24
>
> Thank you for your time and effort reviewing our paper. We respond to your questions and concerns below.
>
> > Q1: Organization of theoretical results
>
> Thanks for this suggestion. In the revised main text, we have carefully structured the presentation so that whenever a theoretical result A is required for establishing another result B (for instance, when A is used in B's proof), A is presented before B. We totally agree that this makes the content more accessible to readers.
>
> Also, to make space for more detailed explanations for our theorems and examples, we have moved several technical details (such as the Intuition before Theorems 1 and 2) to Appendix.
>
> > Q2: The font is somewhere inconsistent.
>
> Thanks for pointing it out, we have fixed it.
>
> > Q3: More explanations for the examples.
>
> Thanks for this advice. We totally agree that more explanations for the examples can help readers better understand our algorithm. In the revised manuscript, we provided more detailed explanations for the examples at the end of Section 3 and Section 4, where we demonstrate the result of our algorithm at each step.
>
> > Q4: Source code
>
> Our source code can be accessed through this link. https://anonymous.4open.science/r/f15G8O29BB

---

> ### Author Response · Authors · 2024-11-25
> **Further Discussion**
>
> Dear Reviewer B7bR:
>
> We really appreciate your constructive opinions that helped us improve this paper. t is really important to us that you could kindly read our rebuttal and provide further questions if there are any.
>
> Thank you so much and hope you have a good day.
>
> Best,
> Authors

---

> ### Comment · Reviewer_B7bR · 2024-11-26
>
> Thank you for the authors' rebuttal. I stand by my original rating.

---

> > ### Author Response · Authors · 2024-11-27
> >
> > Dear Reviewer B7bR:
> >
> > Thanks for your response. We are grateful for your time and labor in helping us improve our manuscript.
> >
> > Best,\
> > Authors

---

### Official Review · Reviewer_BthH · 2024-11-03

**Soundness:** 2
**Presentation:** 3
**Contribution:** 3
**Rating:** 6
**Confidence:** 4

**Summary:**

This paper proposes a causal discovery method with latent variables. Instead of assuming that each latent variable has multiple pure children, they assume that each latent variable has at least one homologous surrogate. They define homologous surrogates and develop identifiability analysis under the homologous-surrogate assumptions. Experiments show the efficacy of their proposed algorithm.

**Strengths:**

1)	The authors tackle a significant but challenging causal discovery problem, that is, identifying latent variables and recovering causal relations between all latent and observed variables.

2)	They introduce two assumptions about the defined homologous surrogates, based on which they develop partial and full identifiability theories. They also propose an algorithm to uncover the causal relationships between latent and observed variables.

**Weaknesses:**

1. The algorithm’s complexity is high, i.e., $O(|O|^4)$ and $O(|O|^2 |L|^2)$ for Algorithms 1 and 2, respectively, which could limit its real-world application. It might be better to give empirical complexity analysis in the experiments, compared with other baselines.

2. Regarding the simulated causal graphs in Figure 5 for the experiments, I think the networks are all small. It could help to investigate the performances when there are more latent variables and when the causal relations between latent variables are sparse.

3. It might be better to give simple descriptions and basic conclusions of the experiment on the real-world dataset Holzinger and Swineford, in the main paper. BTW, from Appendix C, the performances on such a real-world dataset look not satisfactory.

4. It is good to give examples for the definitions. But I think giving examples for the theorems also matters since it is hard to catch the main idea or intuitions for the basic theorems.

**Questions:**

1.	In Equation (7), how are the parents $O_j$ of $O_i$ identified? $O_j$ here might be observed ancestors of $O_i$? It is not clear which step aims to find the parents. Since $M_O^O$ is derived from Algorithm 1, which could contain some spurious edges.

2.	As illustrated by the authors, their algorithms suffer from cases with small sample sizes but could achieve ideal performances when assumptions are met and the sample sizes are large enough. For example, in ideal Case 2 of the experiments, how many samples are needed to achieve the ideal results?

3.	If the number of latent variables is not estimated correctly, what would happen? Did it greatly impact the final results?

4)	The following paper [1] tackles the same problem and also allows non-pure children. Hence I suggest discussing the differences and connections with it.

[1] Xie F, Huang B, Chen Z, et al. Generalized independent noise condition for estimating causal structure with latent variables[J]. Journal of Machine Learning Research, 2024, 25: 1-61.

---

> ### Author Response · Authors · 2024-11-24
>
> # Part (1/2)
>
> Thank you for your careful reading and assessment of our work as well as the suggestions. We respond to your questions below.
>
> > Q1: More complex simulated causal graphs
>
> We totally agree that it is helpful to investigate the case with more latent variables with sparse causal relations. In the revised manuscript, we used simulated causal graphs whose latent causal structure are $L _1 \leftarrow L _2 \to L _3 \to L _4 \to L _5$, whereas we used those with only three mutually adjacent latent variables previously.
>
> > Q2: Empirical complexity analysis
>
> We have reported the running time of each algorithm, which is also summarized here. Our algorithm is far more efficient than both LaHME and PO-LiNGAM, which is expected because LaHME has factorial complexity w.r.t. the number of variables in the worse case (as explicitly mentioned in Section 3.3 in [1]) while PO-LiNGAM has exponential time complexity in the worst case (as explicitly mentioned in Appendix E in [2]).
>
> | Case   | Method    | 2k           | 5k           | 10k          |
> |--------|-----------|--------------|--------------|--------------|
> | Case 1 | GIN       | 1.56±0.13| 1.74±0.14| 2.07±0.18|
> |        | LaHME     | 2.01±0.14    | 2.32±0.25    | 2.96±0.23    |
> |        | PO-LiNGAM | 121.62±27.26 | 116.31±27.64 | 117.39±16.61 |
> |        | Ours      | 2.64±0.27    | 3.05±0.27    | 4.01±0.08    |
> | Case 2 | GIN       | 3.13±0.31| 3.47±0.33| 4.13±0.44|
> |        | LaHME     | 36.60±14.43  | 86.45±71.08  | 116.15±96.61 |
> |        | PO-LiNGAM | 2214.19±779.92| 2073.97±678.85| 2482.57±704.83|
> |        | Ours      | 6.27±0.54    | 7.72±0.33    | 9.69±0.72    |
> | Case 3 | GIN       | 1.64±0.12| 1.79±0.11| 2.22±0.11|
> |        | LaHME     | 6.36±1.18    | 8.64±1.13    | 10.26±1.19   |
> |        | PO-LiNGAM | 295.43±79.55 | 289.59±40.06 | 369.12±31.68 |
> |        | Ours      | 2.58±0.34    | 3.23±0.28    | 4.36±0.38    |
> | Case 4 | GIN       | 0.79±0.07| 0.87±0.08| 1.05±0.08|
> |        | LaHME     | 11.88±1.48   | 13.61±1.67   | 17.51±0.95   |
> |        | PO-LiNGAM | 63.39±16.79  | 69.07±17.81  | 89.61±26.58  |
> |        | Ours      | 1.68±0.28    | 1.85±0.11    | 2.30±0.08    |
>
> > Q3: Result on real-world dataset.
>
> We have explained our result on HS1939 dataset in the revised main text. "Our algorithm correctly identifies the textual factor while merges the visual factor and the speed factor into a single factor. This can be attributed to the fact that both the visual factor and speed factor depends on innate abilities, while the textual factor highly depends on learning experience."
>
> > Q4: Examples for theorems.
>
> Thanks for this suggestion. We have added both intuitive explanations and examples for theorems to make it easier for readers to understand. Taking Theorem 1 as an example, its Intuition states that "The part before 'if and only if' means that all ancestors of $O _i$ are in $\mathbf{J} \cup \mathbf{K}$, that is, $O _i$ is a root variable among $\mathbf{V} \backslash (\mathbf{J} \cup \mathbf{K})$; the part after 'if and only if' means that $O _i$ satisfies certain independence constraints. Therefore, this theorem provides a method for identifying observed root variables via statistical analysis." Also, we provide an example. "Suppose the underlying causal graph is shown as Figure 1. Initially, $\mathbf{J} = \mathbf{K} = \emptyset$. We can identify $O _1$ as an observed root because $\forall O _j \in \\{O _2, ..., O _{11}\\}, \mathrm{R}(O _j, O _1 | \tilde{O} _1) \perp \tilde{O} _1$".
>
> > Q5: How are the parents $O _j$ of $O _i$ identified
>
> First, we would like to clarify the distinction between two matrices $\mathbf{M}$ and $\mathbf{A}$. In our paper, $\mathbf{M}$ is called the mixing matrix while $\mathbf{A}$ is called the adjacency matrix. The entry of $\mathbf{M}$, $m^ {V_ i}_ {V_ j} \neq 0$ implies that $V_ i$ is an ancestor of $V_ j$, whereas the entry of $\mathbf{A}$, $a^ {V_ i}_ {V_ j} \neq 0$ implies that $V_ i$ is a parent of $V_ j$. As you notice, $\mathbf{M}^ {\mathbf{O}}_ {\mathbf{O}}$ is estimated by our partial recovery algorithm. Please note that based on equation (3) in Section 2, we have $\mathbf{M}^ {\mathbf{O}}_ {\mathbf{O}} = (\mathbf{I} - \mathbf{A}^ {\mathbf{O}}_ {\mathbf{O}})^{-1}$ where $\mathbf{I}$ is an identity matrix, so we can derive $\mathbf{A}^ {\mathbf{O}}_ {\mathbf{O}}$ from $\mathbf{M}^ {\mathbf{O}}_ {\mathbf{O}}$. In summary, for observed variables, we can derive their parental relationships from their ancestral relationships.
>
> > Q6: Samples needed for ideal results
>
> Ideally, our algorithm would achieve 0 error in latent variables, 1 correct-ordering rate, and 1 F1-score in the new case 2. With 20k samples, it yielding an 0.2 average error in latent variables, 0.97 average correct-ordering rate, and 0.94 average F1-score, which closely approach the ideal performance. Moreover, we find that with 50k samples, it can achieves ideal results.

---

> ### Author Response · Authors · 2024-11-24
>
> # Part (2/2)
>
> > Q7: What if the number of latent variables is estimated incorrectly
>
> First, we would like to highlight that the number of latent variables need not to be known a priori. Moreover, our algorithm even does not incorporate a step for estimating the number of latent variables. Instead, it directly identifies latent variables and their causal relations without requiring either the true or estimated number of latent variables. Instead, the number of latent variables is directly read off from the final result.
>
> Second, if the algorithm misses any genuine latent variable or introduces any spurious one, the estimated number of latent variables will be incorrect and the final result returned by the algorithm will be locally inconsistent with the ground truth. Specifically, if the algorithm misses a genuine latent variable, causal relations associated with the missed latent variable in the underlying causal graph will not be represented in the result. If the algorithm introduces a spurious latent variable, causal relations associated with the introduced latent variable in the result cannot find their correspondences in the underlying causal graph.
>
> > Q8: Connections and differences between our work and [3].
>
> Thanks for bringing this recent work to our attention. In the following, we review works on causal discovery with latent variables and explicitly position our work and [3] in their appropriate context. This part is added to the revised manuscript.
>
> Works on causal discovery with latent variables can be classified into three categories. The first category assumes that latent variables are mutually independent. The second category allows the presence of causally-related latent variables but cannot identify latent variables, let alone their causal relations (An example of the second category is FCI). The third category not only allows the presence of causally-related latent variables but also can identify latent variables and their causal relations. **Both our work and [3] belongs to the third category**.
>
> Recent works in the third category predominantly rely on the pure children assumption that latent variables have pure children. They not only identify latent variables by locating their pure children but also use their pure children as proxies to infer their causal relations. These works can be further categorized into two groups. Some works make the special pure children assumption that each latent variable has multiple pure children. Here, a variable is said a pure child of anther only if the latter is the only parent of the former. Other works make the general pure children assumption that each latent variable belongs to a latent set (comprising one or more latent variables) which has sufficient pure children. Here, a variable is said a pure child of a latent set only if all parents of the former are within the latter. It should be noted that the weaker general pure children assumption is often accompanied by local unidentifiability: for multiple latent variables within a latent set, even the existence (let alone directions) of the causal relations between them might be undeterminable. **[3] makes the general pure children assumption**, where pure children can be either latent or observed variables. Specifically, in that paper, the word 'pure children' explicitly appears in Definition 8 while Definition 8 explicitly appears in Condition 1.
>
> In contrast to the above works including [3], we make neither the special pure children assumption nor the general pure children assumption. By introducing the concept of homologous surrogate, we eliminate the need for pure children. The homologous surrogate fundamentally differs from the pure child in the sense that the former allows for much more flexible parents. For instance, if an observed variable $O$ is a pure child of a latent variable $L$, $O$ must have only one parent $L$; but if $O$ is a homologous surrogate of $L$, $O$ is allowed to have other parents besides $L$, such as other latent parents provided that they are all $L$'s ancestors.
>
> In summary, [3] and our work broaden the applicability of causal discovery from different perspectives, neither work diminishes the contributions of the other.
>
> **Reference**
>
> [1] Xie F, Huang B, Chen Z, et al. Identification of linear non-gaussian latent hierarchical structure. ICML 2022.
>
> [2] Jin S, Xie F, Chen G, et al. Structural estimation of partially observed linear non-gaussian acyclic model: A practical approach with identifiability. ICLR 2024.
>
> [3] Xie F, Huang B, Chen Z, et al. Generalized independent noise condition for estimating causal structure with latent variables. Journal of Machine Learning Research 2024.

---

> ### Author Response · Authors · 2024-11-25
> **Further Discussion**
>
> Dear Reviewer BthH:
>
> We thank you again for your careful reading and assessment of our work. We have taken our maximum effort to address your every concern. If there are any concerns unresolved, we would be glad to have further discussions.
>
> Thanks again for your time, looking forward to hearing from you soon.
>
> Best,
> Authors.

---

> ### Author Response · Authors · 2024-12-02
>
> Dear Reviewer BthH:
>
> Thanks again for your valuable time and effort in reviewing our paper. As the discussion period approaches its end, we would like to confirm whether our rebuttal has adequately addressed all your concerns. If you have any remaining question or require further clarification, we are glad to provide explanations. Also, we would greatly appreciate any suggestions you might have for further improving the quality and presentation of our paper.
>
> Sincerely,\
> Authors

---

### Official Review · Reviewer_AU92 · 2024-11-04

**Soundness:** 3
**Presentation:** 2
**Contribution:** 2
**Rating:** 6
**Confidence:** 3

**Summary:**

This paper proposes a method for causal discovery with latent variables while relaxing the common assumption that requires the existence of pure children. Instead, the method relies on the condition of homologous surrogates, which can have multiple parents. The proposed algorithms are validated on synthetic data.

**Strengths:**

1. The theoretical results are clearly stated with all definitions introduced clearly.

2. The figures are helpful in understanding the method.

3. Experiments in various settings validate the effectiveness of the algorithm.

4. The limitations have been discussed explicitly.

**Weaknesses:**

1. Since "homologous surrogate" is a newly introduced term, it would be helpful to provide a concrete example immediately when it is first mentioned in the manuscript. This would aid readers in understanding the concept more intuitively.

2. Some of the most closely related works have not been mentioned in the introduction. For example, [Dong et al., 2024] consider almost the same problem with a very similar motivation, i.e., relaxing the structural assumptions among latent and observed variables. That work also allows the existence of children for observed variables. Currently, the discussion was only in the remark and related work at the end of the manuscript, but not in the introduction.

3. In line 82, it states that the considered setting is new in causal discovery. However, it might not be accurate since causally-related latent variables have been considered before, even in the presence of observed variables' children. I understand that omitting works such as [Dong et al. 2024] in the introduction would make the story clearer, but it might introduce confusion for readers less familiar with the field.

4. In line 45, it seems to state that [Jin et al. 2024] make the pure children assumption, which might also not be true.

**Questions:**

Please refer to the comments above. Some statements in the current manuscript are not precise, and the overlooking of some closely related work in the introduction may introduce confusion about the unique contributions of this work.

---

> ### Author Response · Authors · 2024-11-24
>
> # Part (1/2)
>
> Thank you for your time and effort reviewing our paper. Before addressing your concerns point by point, we would like to first review previous works on causal discovery with latent variables and highlight the key distinction between our work and previous research. This part can be found in the revised Introduction.
>
> ## Related Works
>
> Works on causal discovery with latent variables can be classified into three categories. The first category assume that latent variables are mutually independent. The second category allow the presence of causally-related latent variables but cannot identify latent variables, let alone their causal relations. The third category not only allows the presence of causally-related latent variables but also can identify latent variables along with their causal relations. Our work belongs to the third category.
>
> Recent works in the third category predominantly rely on the pure children assumption that latent variables have pure children. They not only identify latent variables by locating their pure children but also use their pure children as proxies to infer their causal relations. These works can be further categorized into two groups. Some works make the special pure children assumption that each latent variable has multiple pure children. Here, a variable is said a pure child of anther only if the latter is the only parent of the former. Other works make the general pure children assumption that each latent variable belongs to a latent set (comprising one or more latent variables) which has sufficient pure children. Here, a variable is said a pure child of a latent set only if all parents of the former are within the latter. It should be noted that the weaker general pure children assumption is often accompanied by local unidentifiability: for multiple latent variables within a latent set, even the existence (let alone directions) of the causal relations between them might be undeterminable. In summary, the concept of pure children is characterized by having strictly restricted parents.
>
> ## Key distinction between our work and previous research
>
> In contrast to the above works, we make neither the special pure children assumption nor the general pure children assumption. In this paper, by introducing the concept of homologous surrogate, we eliminate the need for pure children. The homologous surrogate fundamentally differs from the pure child in the sense that the former allows for much more flexible parents. For instance, if an observed variable $O$ is a pure child of a latent variable $L$, $O$ must have only one parent $L$; but if $O$ is a homologous surrogate of $L$, $O$ is allowed to have other parents besides $L$, such as other latent parents provided that they are all $L$'s ancestors.

---

> ### Author Response · Authors · 2024-11-24
>
> # Part (2/2)
>
> > Q1: Concrete example for homologous surrogate
>
> Thanks for this suggestion. We totally agree that providing a concrete example when first mentioning homologous surrogate helps readers to develop intuitive understanding. In the revised manuscript, we have added the context "For instance, if an observed variable $O$ is a homologous surrogate of a latent variable $L$, $O$ is allowed to have other parents besides $L$, such as other latent parents provided that they are all $L$'s ancestors. Taking Fig. 1 as an example, although $O _3$ has two parents $L _1, L _2$, it can still serve as a homologous surrogate of $L _2$." immediately after first mentioning homologous surrogate.
>
> ---
>
> > Q2: [1] relaxes structural assumptions.
>
> We fully acknowledge the significant contributions of [1]. Specifically, it not only allows ordinary observed variables to have children, but also allow an observed variable which serves as a pure child of a latent variable to have children of its own. We emphasize this point in the revised Introduction.
>
> On the other hand, we would like to highlight that [1] does not diminish our contribution, because [1] makes the general pure children assumption while we eliminate the need for pure children via homologous surrogates which is fundamentally different from pure children as stated above.
>
> ---
>
> > Q3: Novelty of our problem setting
>
> We would like to highlight that the key feature of our problem setting lies in its introduction of homologous surrogates, moving beyond the conventional pure children framework. Although both [1] and our work can handle causally-related latent variables, [1] makes the general pure children assumption whereas we do not. We have emphasized this point in the revised manuscript.
>
> ---
>
> > Q4: Whether [2] makes the pure children assumption
>
> [2] makes the general pure children assumption. In Abstract of [2], the authors explicitly state that "We theoretically show that with the aid of high-order statistics, the causal graph is (almost) fully identifiable if, roughly speaking, each latent set has a sufficient number of pure children."
>
> ---
>
> Finally, we would like to highlight again that [1,2] and our work broaden the applicability of causal discovery from different perspectives, none of which diminishes the contributions of others. Specifically, [1,2] substantially relax the structural constraints under the general pure children assumption, e.g., they allow pure children to be latent variables and allow observed pure children to have children of their own. Different from them, we investigate a new problem setting that introduces homologous surrogates, moving beyond the conventional pure children framework. In this new setting, we provide both solid theoretical results and novel causal discovery algorithms.
>
> ---
>
> **Reference**
>
> [1] Dong X, Huang B, Ng I, et al. A versatile causal discovery framework to allow causally-related hidden variables. ICLR 2024.
>
> [2] Jin S, Xie F, Chen G, et al. Structural estimation of partially observed linear non-gaussian acyclic model: A practical approach with identifiability. ICLR 2024.

---

> ### Author Response · Authors · 2024-11-25
> **Further Discussion**
>
> Dear Reviewer AU92:
>
> We want to express our appreciation for your valuable suggestions, which greatly helped us improve the quality of this paper. We have taken our maximum effort to address your concerns on clarification.
>
> Your further opinions are very important for evaluating our revised paper and we are hoping to hear from you. Thank you so much.
>
> Best,
> Authors.

---

> > ### Comment · Reviewer_AU92 · 2024-11-25
> >
> > Thanks for your responses and updates. The current version, especially the introduction, has been modified a lot to discuss related work from a more comprehensive perspective. New categories like "special" and "general" pure children assumptions have been introduced. My previous concerns regarding the related work have been addressed, and I have adjusted my score accordingly.

---

> > > ### Author Response · Authors · 2024-11-26
> > >
> > > Dear Reviewer AU92:
> > >
> > > Thank you very much for your swift response. It has been our genuine pleasure to provide clarification and address each of your concerns comprehensively.
> > >
> > > Best,\
> > > Authors

---

### Official Review · Reviewer_uc12 · 2024-11-04

**Soundness:** 2
**Presentation:** 2
**Contribution:** 2
**Rating:** 6
**Confidence:** 4

**Summary:**

This paper introduces the concept of homologous surrogates to address the challenges associated with causal discovery involving latent variables. It formulates two assumptions regarding homologous surrogates and develops theoretical results based on each assumption. Under the weaker assumption, the authors demonstrate the feasibility of partially recovering the causal graph by identifying the ancestors of each variable. The stronger assumption facilitates the exact identification of each variable's parents, enabling full recovery of the causal graph. Additionally, the paper proposes an algorithm that effectively leverages the properties of homologous surrogates for causal graph recovery and validates its effectiveness through experimental results.

**Strengths:**

1. This paper deals with a causal structure learning problem involving latent variables, which is significant for the field of causal discovery.

2. The authors proposes a method for identify the causal structure via the homologous surrogates under mild assumptions.

**Weaknesses:**

1. The representation needs refinement, as the content of the paper lacks logical flow and coherence, which makes it difficult for readers to follow the main points.

2. The technical contributions appear to be somewhat constrained. This paper draws upon two established methodologies, pseudo-residuals and causal effect estimation via cumulants. It introduces the concept of homologous children, which is fundamentally equivalent to the notion of pure children.

3. The authors mentioned that their work expands the applicability of causal discovery methods, and that homologous surrogates aid in causal discovery with latent variables. However, in the experimental results involving real-world data, their proposed method performs poorly. The authors believe this is due to a limited sample size. So, are there more suitable real-world datasets available to validate the method proposed in this paper?

**Questions:**

See the weaknesses above.

---

> ### Author Response · Authors · 2024-11-24
>
> # Part (1/3)
>
> Thank you for your assessment of our work. We address each of your concerns point by point below.
>
> > Q1: logical flow and coherence
>
> Thanks for pointing this out. To enhance logical flow and coherence of our paper, we have tried our best to improve the presentation, especially Section 1 (Introduction), Section 3 (Partial Identification), and Section 4 (Full Identification).
>
> ## Section 1
>
> We have refined the Introduction to give readers a clearer panoramic view of our work. For instance, to help readers understand how our work is positioned in the field of causal discovery, we have made the following efforts.
>
> In the first paragraph, we provide a well-structured review of previous works on causal discovery with latent variables and position our work in appropriate context. Specifically, these works can be classified into three categories. The first category assume that latent variables are mutually independent. The second category allow the presence of causally-related latent variables but cannot identify latent variables, let alone their causal relations. The third category not only allows the presence of causally-related latent variables but also can identify latent variables along with their causal relations. Our work belongs to the third category.
>
> In the second paragraph, we provide a comprehensive review of studies falling within the same category as our work. Specifically, these works predominately rely on the pure children assumption that latent variables have pure children, which not only identify latent variables by locating their pure children but also use their pure children as proxies to infer their causal relations. The concept of pure children is characterized by having strictly restricted parents. For instance, a variable is said a pure child of anther only if the latter is the only parent of the former.
>
> After clearly delineating the related works, in the third paragraph, we highlight the key distinction between our work and previous research. Specifically, we eliminate the need for pure children by introducing the concept of homologous surrogate. The homologous surrogate fundamentally differs from the pure child in the sense that the former allows for much more flexible parents. For instance, if an observed variable $O$ is a pure child of a latent variable $L$, $O$ must have only one parent $L$; but if $O$ is a homologous surrogate of $L$, $O$ is allowed to have other parents besides $L$, which can be either latent or observed.
>
>
> ## Section 3&4
>
> The main contributions of our work are detailed in Section 3 and Section 4. Taking Section 3 as an example, we have made the following efforts to improve its readability.
>
> - At the outset of Section 3, we give the formal definition of homologous surrogate and the corresponding assumption, accompanied by concrete examples to enhance understanding. Furthermore, we also provide a detailed explanation of how they differ from related concepts (such as pure child) and assumptions (such as pure children assumption) that appeared in previous literature.
>
> - Next, we organize the technical details following the flow of our proposed algorithm. First, we give a high-level overview of our algorithm, where we enumerate key algorithmic steps in sequence and also link each step to its corresponding theoretical foundation. In the following, using each key algorithmic step as a subsection header, we present the corresponding implementation and theoretical foundation in precise mathematical terms. Finally, we summarize our algorithm in Algorithm 1 and also give an example in Figure 1 to illustrate how the algorithm operates. In the revised manuscript, the example is explained in more details.
>
> - Considering that some theorem statements are necessarily lengthy to ensure rigor, we understand readers may find it challenging to grasp the intuition. To address this, we provide "Intuition" and "Example" following each theorem to help readers better understand its implication. Taking Theorem 1 as an example, its Intuition states that "The part before 'if and only if' means that all ancestors of $O _i$ are in $\mathbf{J} \cup \mathbf{K}$, that is, $O _i$ is a root variable among $\mathbf{V} \backslash (\mathbf{J} \cup \mathbf{K})$; the part after 'if and only if' means that $O _i$ satisfies certain independence constraints. Therefore, this theorem provides a method for identifying observed root variables via statistical analysis." Also, we provide an example. "Suppose the underlying causal graph is shown as Fig. 1. Initially, $\mathbf{J} = \mathbf{K} = \emptyset$. We can identify $O _1$ as an observed root because $\forall O _j \in \\{O _2, ..., O _{11}\\}, \mathrm{R}(O _j, O _1 | \tilde{O} _1) \perp \tilde{O} _1$."
>
> Please let us know if you still have any concern or suggestion for further improvement. We are fully committed to enhancing the quality of our manuscript and will make every effort to address your feedback.

---

> ### Author Response · Authors · 2024-11-24
>
> # Part (2/3)
>
> > Q2: Technical contributions
>
> First, we summarize our major technical contributions as follows. Particularly, we would like to highlight that our major technical contributions lie in development of systematic theoretical results and algorithms for a previously unexplored problem setting rather than specific implementation details.
>
> - We introduce a new concept called homologous surrogates, which is fundamentally different from conventional pure children in the sense that the latter is characterized by having strictly restricted parents while the former allows for much more flexible parents
>
> - We formulate two assumptions involving homologous surrogates, and we prove that the causal graph can be partially/fully recovered under the weaker/stronger assumption.
>
> - We propose algorithms that fully exploit the properties of homologous surrogates to partially/fully recover the causal graph under the weaker/stronger assumption.
>
> Second, we address your specific concerns about homologous surrogates, cumulants, and pseudo-residual.
>
> - **Homologous surrogates.** At a high level, homologous surrogate is fundamentally different from the pure child in the sense that the latter is characterized by having strictly restricted parents while the former allows for much more flexible parents. More details are as follows.
>   * In terms of definitions, although the precise definition of pure children varies across different works, there is a consensus that if a variable is called a pure child of another, the former must have no other parent except the latter. In contrast, according to the definition of homologous surrogate (Definition 1 in our paper), if a variable is called a homologous surrogate of another, the former is totally permitted to have other parents besides the latter, and the other parents could be either latent variables or observed variables.
>   * In terms of algorithms, existing causal discovery algorithms based on the pure children assumption identify a latent variable by locating its multiple pure children, which cannot handle a causal graph where each latent variable has only one homologous surrogate rather than multiple pure children.
>
> - **Cumulants.** In our paper, cumulants are employed to construct Corollary 3. Because cumulant is a fundamental and commonly-used statistical measure, the use of it neither constitutes a novel contribution nor diminishes our other contributions. We elaborate how fundamental it is and how it is widely used below.
>   * Cumulant is a very fundamental concept in statistics, covariance is simply a second-order cumulant. Given two random variables $X _i, X _j$, $\mathrm{Cov}(X _i, X _j) = \mathrm{Cum}(X _i, X _j)$.
>   * While the widespread use of covariance is indisputable, the fourth-order cumulant has been applied in signal processing since the last century, especially in the topic of independent component analysis (ICA) [1,2]. The mixing coefficients in ICA are equivalent to the accumulated causal effects in linear latent non-Gaussian model.
>
> - **Pseudo-residuals.** We respectfully disagree that the use of pseudo-residuals weakens our contributions. The reasons are as follows.
>   * As mentioned in Intuition of Definition 2, pseudo-residual proposed in [3] is a simple variant of conventional residual. That is, rather than being a well-developed method that can be directly applied to solve causal discovery problems, pseudo-residual is merely a basic module. To transform this basic module into a well-develop method that can solve our problem, we have to make many innovative designs.
>   * The theoretical results based on pseudo-residual in our paper are fundamentally different from those in [3]. Specifically, with the pure children assumption, theorem 2 in [3] provides a sufficient and necessary condition for two observed variables $O_i, O_j$ to be pure children of a same latent (not necessarily root) variable: for any other $O_k$, $R(O_i, O_j|O_k) \perp O_k$. In contrast, with the homologous surrogates assumption, the former only provides a necessary but not sufficient condition for a single observed variables $O_i$ to be a homologous surrogate of a latent root variable: for any other $O_j$ and $O_k$, $R(O_j, O_k | \tilde{O}_i) \perp \tilde{O}_i$.

---

> ### Author Response · Authors · 2024-11-24
>
> # Part (3/3)
>
> > Q3: Our work expands the applicability of causal discovery methods
>
> First, we claim that our work expands the applicability of causal discovery methods because it can handle a previously unexplored problem setting that involves homologous surrogates, moving beyond the conventional pure children framework. Our main contributions lie in algorithmic innovation and theoretical advancement: we propose a novel algorithm for this new setting, and more importantly, establish a solid theoretical framework that provides a complete characterization of the algorithm's behavior, eventually proving its asymptotic correctness. Our empirical results on synthetic datasets strongly support our theoretical analysis, showing excellent consistency between theoretical results and experimental results when the sample size is sufficiently large.
>
> Second, we acknowledge that our algorithm has limitations in handling cases with limited sample sizes. This is primarily attributed to the fact that our algorithm relies on the estimation of high-order cumulants, which usually exhibits large estimation errors under limited sample sizes. Moreover, our algorithm operates in a progressive manner, of which each step builds upon the previous one, so errors are propagated and amplified during this process. Therefore, in real-world dataset HS1939 consisting of only 301 samples, its returned result is not completely consistent with that provided by human experts. For most research fields except computer science, non-professionals typically have no access to large-scale datasets.
>
> Finally, we suggest that the value and potential impact of a causal discovery algorithm should be assessed from a long-term perspective. Many widely-used algorithms began as theoretical frameworks, with their practical performance improving over time through technical refinements. For instance, when first proposed in [4], FCI algorithm is theoretically elegant, but its initial practical performance was limited by computational complexity and stability issues. Subsequent works gradually improved its practical performance through various modifications like RFCI [5] for better computational efficiency, and FCI+ [6] for enhanced stability and accuracy.
>
>
>
> **Reference**
>
> [1] Thi H L N, Jutten C. Blind source separation for convolutive mixtures. Signal processing, 1995, 45(2): 209-229.
>
> [2] Hyvärinen A, Oja E. A fast fixed-point algorithm for independent component analysis. Neural computation 1997.
>
> [3] Cai R, Xie F, Glymour C, et al. Triad constraints for learning causal structure of latent variables. NeurIPS 2019.
>
> [4] Spirtes P L, Meek C, Richardson T S. Causal inference in the presence of latent variables and selection bias. UAI 1995.
>
> [5] Colombo D, Maathuis M H, Kalisch M, et al. Learning high-dimensional directed acyclic graphs with latent and selection variables. The Annals of Statistics 2012.
>
> [6] Claassen T, Mooij J, Heskes T. Learning sparse causal models is not NP-hard. UAI 2013.

---

> ### Author Response · Authors · 2024-11-25
> **Further Discussion**
>
> Dear Reviewer uc12,
>
> We really appreciate your efforts to help improve this paper. We have carefully addressed your concerns. It is really important to us that you could kindly read our rebuttal and provide further questions if there are any.
>
> Thank you so much and hope you have a good day.
>
> Best,
> Authors.

---

> > ### Comment · Reviewer_uc12 · 2024-11-26
> >
> > Thanks a lot for the authors' detailed response, which addressed most of my concerns. I’m happy to increase my evaluation score.

---

> > > ### Author Response · Authors · 2024-11-26
> > >
> > > Dear Reviewer uc12:
> > >
> > > We would like to express our deep gratitude for your prompt feedback, and we are delighted to have addressed your concerns.
> > >
> > > Sincerely,\
> > > Authors

---

### Author Response · Authors · 2024-11-25
**General Response**

We sincerely thank all reviewers for their insightful and valuable feedback on our manuscript. Their constructive comments have helped us significantly improve the quality of our work. We have carefully addressed each concern raised and made substantial improvements to the manuscript. The major revisions are summarized below.

We have substantially refined the presentation of our manuscript, especially in Sections 1, 3, and 4, to improve logical flow and coherence. An an example, we discuss this improvement in detail in our response to Q1 of Reviewer uc12

We have provided more detailed discussions on the connections and differences between our work and previous research in Introduction. As an example, we discuss this improvement in detail in our response to Reviewer AU92.

We have provided intuitive explanations and illustrative examples for key theorems. Besides, we have also added more detailed explanations for the illustrative examples of our proposed algorithm. As examples, we discuss this improvement in detail in our response to Q4 of Reviewer BthH and Q3 of Reviewer B7bR.

We have provided more detailed experimental results on more complex causal graphs. As examples, we discuss this improvement in detail in our response to Q1 and Q2 of Reviewer BthH.

We have open-sourced our code following the suggestion of Reviewer B7bR. https://anonymous.4open.science/r/f15G8O29BB

Thank you again for your time and expertise in reviewing our work.

---

### Author Response · Authors · 2024-12-02
**Rebuttal Summary**

Dear reviewers, AC, and SAC:

We sincerely thank all reviewers for taking their valuable time to read our paper, provide constructive suggestions, and actively engage in discussions. We are also very grateful to AC and SAC for organizing high-quality review process. In the last couple of weeks, we have tried our best to address each concern of all reviewers and integrated these changes into our revised manuscript. The main revisions are summarized as follows.

1. To help readers better understand the position of our work in the field of causal discovery, we have added the following content to the Introduction section: a well-structured review of previous research on causal discovery with latent variables, along with a comprehensive analysis comparing our work with closely related studies that rely on the pure children assumption.

2. To help readers better comprehend our theoretical result, we provide both intuitive explanations and illustrative examples for the key theoretical results such as Theorems 1, 2, 3.

3. To help readers better follow our algorithm, we have provided more detailed explanations at the end of Sections 3 and 4 for the running examples presented in Figures 3 and 4. Besides, we have provided more comprehensive experimental results shown as Table 1 on more complex causal graphs shown as Figure 5. Finally, we have released our source code through an anonymous link (https://anonymous.4open.science/r/f15G8O29BB).

We believe these revisions have substantially improved our paper, and we will continue to refine our paper in the future. Also, we would like to highlight the main contributions of our paper below.

1. We propose a new concept called homologous surrogate, which fundamentally differs from the conventional pure children in the sense that the latter is characterized by having strictly restricted parents while the former allows for much more flexible parents.

2. We formulate two assumptions involving homologous surrogates and develop novel theoretical results under each assumption. These theoretical results imply that the causal graph can be partially/fully recovered under the weaker/stronger assumption.

3. Building on our theoretical results, we derive a systematic and innovative algorithm which fully leverages the properties of homologous surrogates for causal graph recovery.

We sincerely hope our work could contribute to the community and advance the development of causal discovery. Thanks again for your efforts.

Sincerely,\
Submission 217 Authors.

---

### Meta-Review · Area_Chair_4n4o · 2024-12-07

**Metareview:**

This paper introduces the concept of homologous surrogates to address the challenges associated with causal discovery involving latent variables. The paper has developed new theoretical results regarding this new concept, which establish that the causal graph can be partially or fully recovered under some reasonable assumptions regarding the homologous surrogates. Building on the established theoretical results, the paper derives a principled and effective algorithm for causal graph recovery.

Following the rebuttal, all reviewers lean towards accepting this paper. Overall, the rebuttal has been effective in addressing key concerns raised by the reviewers, as summarized below.

Reviewer uc12 raised concerns regarding the presentation of the technical narratives as well as several clarification questions regarding the novelty of the introduced concept and limitation of the proposed methods in some experiment settings with limited sample sizes. The authors have revised the paper substantially to address concerns regarding clarity. The rebuttal has also acknowledged the limitation of the proposed method in experiments with limited sample sizes. But, this limitation clearly does not negate the contribution here.

Reviewer AU92 raised concerns regarding the lack of examples, discussion of related work, and clarification of the novelty of the newly introduced concepts. The rebuttal has satisfactorily addressed these concern and the reviewer has adjusted the rating accordingly.

Reviewer B7br raised concerns regarding the paper organizations, explanations for the examples, and source code releases. The rebuttal has addressed these points sufficiently, retaining the reviewer's acceptance rating.

Reviewer BthH raised concerns regarding the complexity of the proposed algorithm, lack of experiments with larger network sizes, and lack of simple examples for definitions. The rebuttal has provided extra experiment results and provided empirical complexity analysis. The reviewer has not responded to the authors' follow-up messages but the rating remains positive. I also think the detailed rebuttal has addressed the raised concerns sufficiently.

Given the consensus, I recommend that this paper be accepted for publication.

**Additional Comments On Reviewer Discussion:**

All reviewers have responded positively to the authors' rebuttal. Overall, there is a clear acceptance signal.

---

### Decision · Program_Chairs · 2025-01-22

Accept (Poster)